# Total biosynthesis of the cyclic AMP booster forskolin from *Coleus forskohlii*

Irini Pateraki[1,2]*[†], Johan Andersen-Ranberg[1,2†‡], Niels Bjerg Jensen[3], Sileshi Gizachew Wubshet[4§], Allison Maree Heskes[1,2], Victor Forman[1], Björn Hallström[5], Britta Hamberger[1,2¶], Mohammed Saddik Motawia[1,2], Carl Erik Olsen[1,2], Dan Staerk[4], Jørgen Hansen[3], Birger Lindberg Møller[1,2], Björn Hamberger[1,2¶]

[1]Plant Biochemistry Laboratory, Department of Plant and Environmental Sciences, University of Copenhagen, Copenhagen, Denmark; [2]Center for Synthetic Biology "bioSYNergy", Copenhagen, Denmark; [3]Evolva A/S, Copenhagen, Denmark; [4]Department of Drug Design and Pharmacology, Faculty of Health and Medical Sciences, University of Copenhagen, Copenhagen, Denmark; [5]Science for Life Laboratory, KTH - Royal Institute of Technology, Stockholm, Sweden

*For correspondence: eipa@plen.ku.dk

[†]These authors contributed equally to this work

Present address: [‡]Department of Plant & Microbial Biology, University of California, Berkeley, United States; [§]Nofima AS, Osloveien, Tromsø, Norway; [¶]Department of Biochemistry and Molecular Biology, Michigan State University, East Lansing, United States

**Abstract** Forskolin is a unique structurally complex labdane-type diterpenoid used in the treatment of glaucoma and heart failure based on its activity as a cyclic AMP booster. Commercial production of forskolin relies exclusively on extraction from its only known natural source, the plant *Coleus forskohlii*, in which forskolin accumulates in the root cork. Here, we report the discovery of five cytochrome P450s and two acetyltransferases which catalyze a cascade of reactions converting the forskolin precursor 13$R$-manoyl oxide into forskolin and a diverse array of additional labdane-type diterpenoids. A minimal set of three P450s in combination with a single acetyl transferase was identified that catalyzes the conversion of 13$R$-manoyl oxide into forskolin as demonstrated by transient expression in *Nicotiana benthamiana*. The entire pathway for forskolin production from glucose encompassing expression of nine genes was stably integrated into *Saccharomyces cerevisiae* and afforded forskolin titers of 40 mg/L.

## Introduction

Plants synthesize an impressive diversity of specialized metabolites enabling them to communicate and adapt to environmental challenges (*Mithöfer and Boland, 2012*; *Woldemariam et al., 2011*). Throughout history, humans have benefited from the medicinal properties of many of these phytochemicals (*Hardy et al., 2012*). Specialized plant metabolites and direct derivatives thereof still constitute more than a third of approved pharmaceuticals (*Cragg and Newman, 2013*; *David et al., 2015*). With over 50,000 known structures according to the 'Dictionary of natural products' (http://dnp.chemnetbase.com/), terpenoids are the largest class of plant specialized metabolites and constitute a vast repository of bio-active natural products including many structurally complex compounds (*Pateraki et al., 2015*). Examples of widely used plant-derived terpenoid pharmaceuticals are the anticancer drug paclitaxel (taxol) (*Liu and Khosla, 2010*), the therapeutic ingenol mebutate (picato) that is used for treatment of actinic keratosis (*King et al., 2016*; *Luo et al., 2016*) and artemisinin which is the most efficient treatment against malaria caused by *Plasmodium* parasites (*Graham et al., 2010*; *Paddon and Keasling, 2014*). Traditional chemical synthesis of plant-derived diterpenoid pharmaceuticals remains economically challenging, despite recent examples of elegant strategies mimicking natural routes (*Appendino, 2014*; *Kawamura et al., 2016*; *Yuan et al., 2016*). Extraction from plant biomass and semisynthesis from biotechnologically produced intermediates

**eLife digest** Unlike animals, plants cannot move away from a herbivore or other threats. Instead, they have evolved to produce a vast array of chemical compounds to protect themselves. Some of these compounds are also important to humans, for example, as medicines or fragrances. Plants usually only produce small amounts of these compounds in mixtures with many other compounds, which makes it difficult to purify them. As a result, the methods of purifying the compounds may require huge amounts of plant material, or be expensive and not environmentally friendly. One solution to this would be to genetically engineer microbes like bacteria or yeast to produce the compounds instead. In order to do that, we need to understand exactly which enzymes the plant uses to make each compound and introduce them into suitable microbes.

A compound called forskolin has been used since ancient times in traditional Indian medicine to treat conditions like high blood pressure, asthma and heart complications. Forskolin is found exclusively in the root of a plant called *Coleus forskohlii*, which is native to India and south-east Asia. It is stored inside cells within the bark of the root in structures called oil bodies, which are similar to oil drops. However, it is not known where forskolin is made, or which enzymes are involved.

Pateraki, Andersen-Ranberg et al. set out to uncover how *C. forskohlii* produces this compound. The experiments show that forskolin is produced within the cells that contain the oil bodies. A technique called RNA sequencing was used to identify several genes that are highly active in these cells and encode enzymes that could potentially be involved in producing forskolin. Further experiments demonstrated that these enzymes drive a cascade of chemical reactions that convert a molecule called 13R-manoyl oxide into forskolin. Next, Pateraki, Andersen-Ranberg et al. inserted the genes into yeast cells that could already produce 13R-manoyl oxide, which allowed the yeast to produce relatively high amounts of forskolin.

These findings show that it is possible to identify the genes involved in the production of medicinal compounds in a relatively short amount of time. This knowledge will aid the development of a method that can be used to produce forskolin and other similar compounds on a large scale without needing to harvest *C. forskohlii* plants.

have been approached as alternative strategies (*Graham et al., 2010*; *Paddon et al., 2013*; *Roberts, 2007*). In contrast to recent examples demonstrating complete pathway reconstruction and production of opiate alkaloids in yeast (*Galanie et al., 2015*; *Nakagawa et al., 2016*), engineered total biosynthesis of terpenoid therapeutics—including paclitaxel and ingenol esters—has not yet been achieved. Challenges on the way to achieving this goal include the identification of pathway enzymes in native systems, particularly for those belonging to multi-enzyme families catalyzing the biosynthesis of specialized metabolites in plants, engineering of poorly understood multi-step enzymatic pathways and difficulties encountered in heterologous expression of key enzymes catalyzing monooxygenations critical for diterpenoid biosynthesis (*Pateraki et al., 2015*; *Renault et al., 2014*).

The diterpenoid forskolin is the active hypotensive principle accumulating in the root cork of *Coleus forskohlii* (*Pateraki et al., 2014*), a perennial shrub of the Lamiaceae family, indigenous to India and Southeast Asia with numerous reported applications in traditional medicine (*Alasbahi and Melzig, 2010b*; *Kavitha et al., 2010*). The pharmaceutical properties of forskolin are based on its ability to directly activate the adenylate cyclase enzyme resulting in elevated levels of the second messenger cyclic adenosine monophosphate (cAMP) (*Doseyici et al., 2014*; *Seamon et al., 1981*). Approved applications of forskolin range from alleviation of glaucoma (OcuforsEye drop solutions, Sabinsa, India), treatment of hypertension and heart failure (Colforsin daropate hydrochloride, a water-soluble derivative of forskolin, Nippon Kayaku, Japan) to lipolysis and body weight control (*Godard et al., 2005*; *Kikura et al., 2004*; *Toya et al., 1998*; *Wagh et al., 2012*; *Yoneyama et al., 2002*). Therapeutic opportunities were also suggested in animal tests, where forskolin-induced pigmentation of the skin, increasing protection against UV-associated carcinogenesis (*D'Orazio et al., 2006*). The complex chemical structure of forskolin with a decalin core, characteristic of labdane-type diterpenoids, a tetrahydropyran ring, five oxidized positions and eight chiral centers (*Figure 1A*) represents a challenge for classical organic chemical synthesis, although a key

intermediate for stereoselective total synthesis has been reported (*Ye et al., 2009*). Hence, commercially available forskolin is extracted from *C. forskohlii* roots and purified from a mixture of over 60 structurally related abietane and epoxylabdane diterpenoids with a forskolin content varying from 0.013% to 0.728% of root dry weight (*Alasbahi and Melzig, 2010a*; *Asada et al., 2012*; *Srivastava et al., 2017*). As the demand for forskolin grows, reliable and sustainable commercial production from *C. forskholii* will become unachievable due to low yields, susceptibility of this

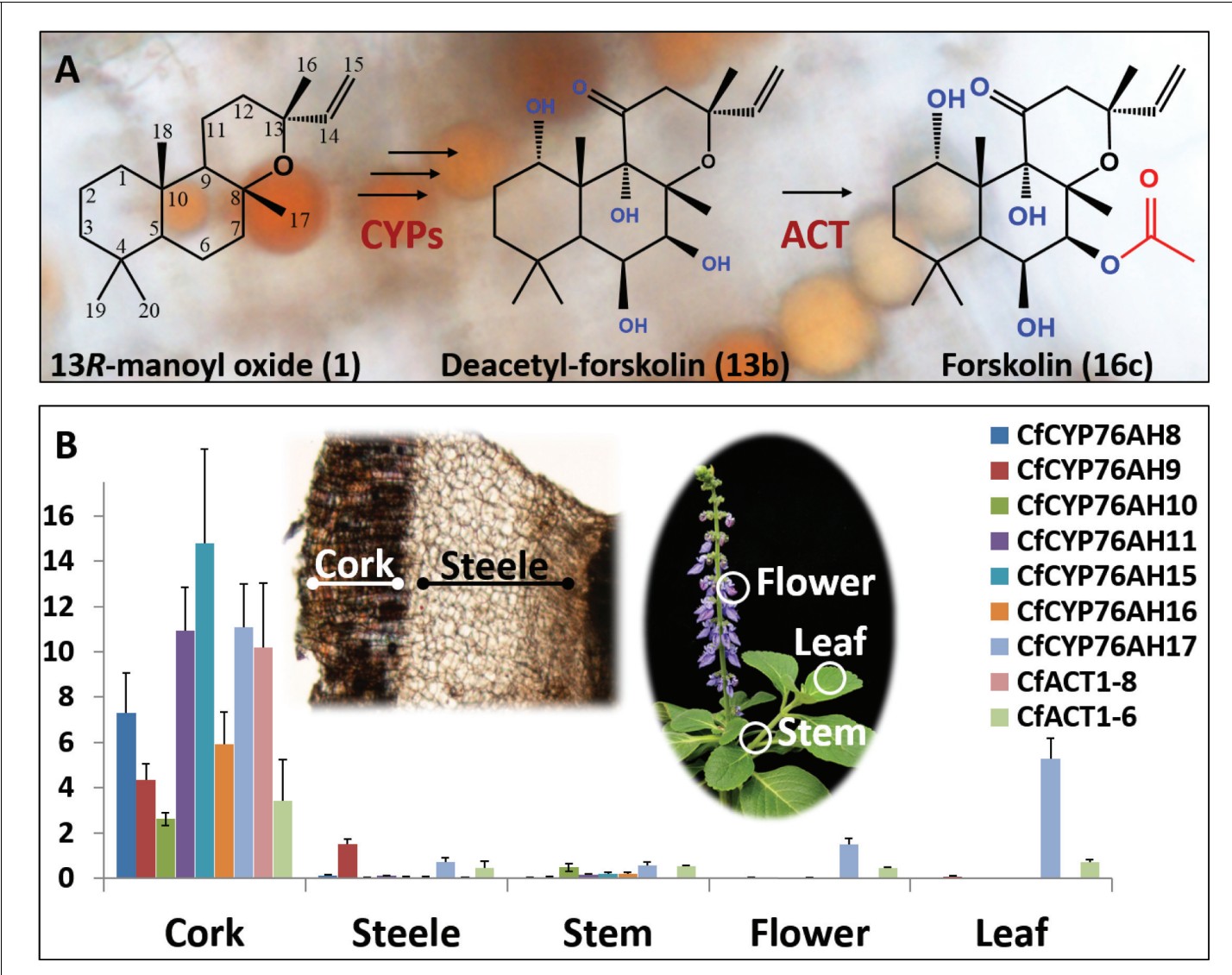

**Figure 1.** Biosynthesis of forskolin in the root cork cells of *C. forskohlii*. (**A**) Scheme showing the structures of the diterpene precursor 13*R*-manoyl oxide, deacetylforskolin and forskolin on a background of root cork cells with forskolin containing oil bodies. (**B**) Transcript profiles of biosynthetic candidate genes in selected tissues of *C. forskohlii* as shown on the illustrations.

The following source data is available for figure 1:

**Source data 1.** cDNAs identified in the *C. forskohlii* root cork transcriptome and cloned during this work, with the GeneBank accession numbers.

**Source data 2.** Table of FPKM (Fragments Per Kilobase of transcript per Million mapped reads) values of the first 20 most abundant cDNAs identified in the root cork transcriptome library.

**Source data 3.** Table of primers used in this study.

species to diseases, changing climatic conditions and the resource intensive extraction and purification procedure required to obtain pharmaceutical grade forskolin (*Mora-Pale et al., 2014*). Elucidation of the biosynthetic pathway to forskolin and subsequent engineering of the pathway into microbial hosts offers a more clear and stable alternative production system that will be better able to address future needs.

Recently, we reported specific accumulation of forskolin and its diterpene scaffold 13*R*-manoyl oxide in the root cork cells of *C. forskohlii*. A pair of diterpene synthases (*Cf*TPS2 and *Cf*TPS3), exclusively present in the root cork, was found to catalyze cyclization of the $C_{20}$ diterpenoid precursor geranylgeranyl diphosphate (GGPP) into 13*R*-manoyl oxide, the diterpene scaffold of forskolin (*Pateraki et al., 2014*). As a proof of concept, biosynthesis of 13*R*-manoyl oxide in enantiomerically pure form but in low yields was achieved by expressing *Cf*TPS2 and *Cf*TPS3 in *Saccharomyces cerevisiae*, *E. coli* and *Synechocystis* sp. (*Andersen-Ranberg et al., 2016*; *Englund et al., 2015*; *Nielsen et al., 2014*). Taking into account the functionalization steps needed for conversion of 13*R*-manoyl oxide to forskolin (*Figure 1A*), involvement of enzymes from the families of cytochrome P450s (CYPs) and acetyltransferases was predicted.

Here, we report an integrated biochemical and functional genomics approach, including metabolomics, single-cell-type transcriptome studies and a synthetic biology modular approach involving transient combinatorial expression of candidate genes in *Nicotiana benthamiana* to identify the panel of enzymes catalyzing functionalization of the *C. forskohlii* diterpene backbones and more specifically the biosynthesis of forskolin. Pathway intermediates were identified using GC- or HPLC-HRMS-SPE-NMR. To demonstrate the downstream application of the present work regarding biotechnological production of forskolin, the entire forskolin biosynthetic pathway was reconstituted in engineered *S. cerevisiae* for fermentation-based production of forskolin from glucose. Forskolin is the first example of a pharmaceutical diterpenoid produced entirely in yeast at titers relevant for industrial consideration. The outlined combinatorial biochemistry approach paves the way for development of yeast-engineered platforms for biosynthesis of other known or new-to-nature diterpenoids

## Results

### Discovery of multifunctional cytochromes P450 in *Coleus forskohlii* producing a multitude of 13*R*-manoyl oxide-derived diterpenoids and identification of a biosynthetic pathway for forskolin

The conversion of 13*R*-manoyl oxide to forskolin requires six regio- and stereospecific monooxygenations and a single regiospecific acetylation (*Figure 1A*). Considering the strict localization of forskolin in the root cork cells of *C. forskohlii* and the almost exclusive expression of the pair of diterpene synthases forming 13*R*-manoyl oxide within the same tissue (*Pateraki et al., 2014*), the root cork was selected for deep RNA-Seq transcriptome analysis. The generated transcriptome contained 263,652 assembled putative cDNAs. The transcriptome was queried for transcripts encoding CYPs belonging to the CYP71 clan, based on their established role in monooxygenation reactions in the biosynthesis of specialized metabolites (*Nelson, 2013*; *Werck-Reichhart and Feyereisen, 2000*). Their relative levels in the root cork transcriptome were also taken into consideration. Within the CYP71 clan, focus was also placed on P450 subfamilies that showed extensive, recent expansions in the cork transcriptome (*Nelson and Werck-Reichhart, 2011*; *Werck-Reichhart and Feyereisen, 2000*). Members of the CYP76AH subfamily, part of the CYP71 clan, have recently been shown to catalyze monooxygenation of abietane-type diterpenoids like miltiradiene and dehydroabietadiene in Lamiaceae species, closely related to *C. forskohlii* (*Božić et al., 2015*; *Ignea et al., 2016a*; *Zi and Peters, 2013*). Members of this P450 subfamily were therefore of high interest as enzymes putatively involved in diterpenoid biosynthesis in *C. forskohlii*. Based on these considerations, a total of 29 CYP candidates (*Figure 1—source data 1*) were selected and cloned in full length using as template cDNA synthesized from root cork total RNA. Among these CYPs, seven members were assigned to the CYP76AH subfamily by the 'P450 Nomenclature committee' (*Nelson, 2009*), rendering this CYP subfamily the highest represented in the transcriptome (*Figure 1—source data 1*). Five were full length sequences (*Cf*CYP76AH8, *Cf*CYP76AH9, *Cf*CYP76AH10, *Cf*CYP76AH11, and *Cf*CYP76AH11), whereas two (*Cf*CYP76AH15 and *Cf*CYP76AH16) were represented by partial cDNAs. For the latter,

5'RACE experiments afforded the full-length cDNAs. Similarly to the previously identified diTPSs, *Cf*TPS2 and *Cf*TPS3 (*Pateraki et al., 2014*), gene expression studies showed that the identified members of the CYP76AH family were highly or exclusively expressed in the root cork cells (*Figure 1B*).

We have recently reported an *Agrobacterium*-mediated modular transient expression system in *N. benthamiana* enabling biosynthesis of labdane-type diterpenes in quantities permitting purification and structural elucidation (*Andersen-Ranberg et al., 2016*). Utilizing this system, all candidate P450 genes were heterologously expressed in combination with genes necessary for the production of high amounts of 13*R*-manoyl oxide (*Andersen-Ranberg et al., 2016*; *Pateraki et al., 2014*). Of the CYPs tested, six efficiently converted 13*R*-manoyl oxide into oxygenated derivatives (*Figures 2A* and *3* and *Figure 3—figure supplements 1* and *2*). *Cf*CYP76AH15, *Cf*CYP76AH8 and *Cf*CYP76AH17 catalyzed formation of 11-oxo-13*R*-manoyl oxide (*2*) as the main product (*Figures 2* and *3*). Forskolin harbors a keto-group at the C-11 position, like the majority of 13*R*-manoyl oxide-derived diterpenoids found in *C. forskohlii* (*Asada et al., 2012*; *Zhang et al., 2009*). Of the three *Cf*CYP76AHs tested in this experiment, *Cf*CYP76AH15 showed the highest efficiency and specificity for the

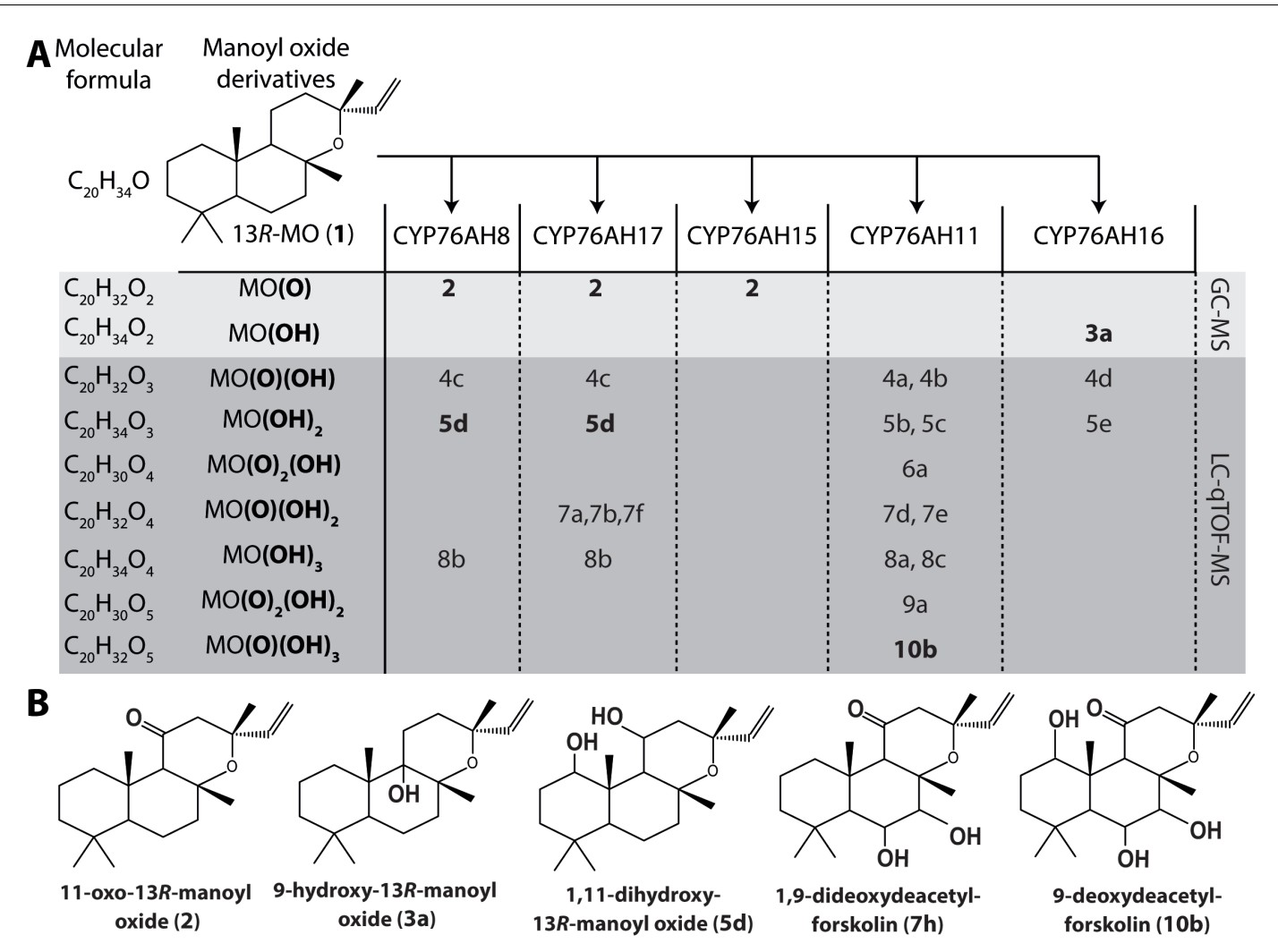

**Figure 2.** 13*R*-manoyl oxide oxide-derived hydroxylated products formed following transient expression of *Cf*CYP76AHs in *N. benthamiana* leaves. (**A**) Molecular formulas of the hydroxylated products obtained using different *Cf*CYP76AHs. The number of hydroxylations of each compound was deduced from its accurate molecular mass (<5 ppm, *Supplementary file 1*) as determined by LC-qTOF-MS or NMR. Each different compound is marked by a number. (**B**) Chemical structures of the compounds marked with numbers in bold in A as determined by NMR (*Tables 1* and *2*). MO: manoyl oxide

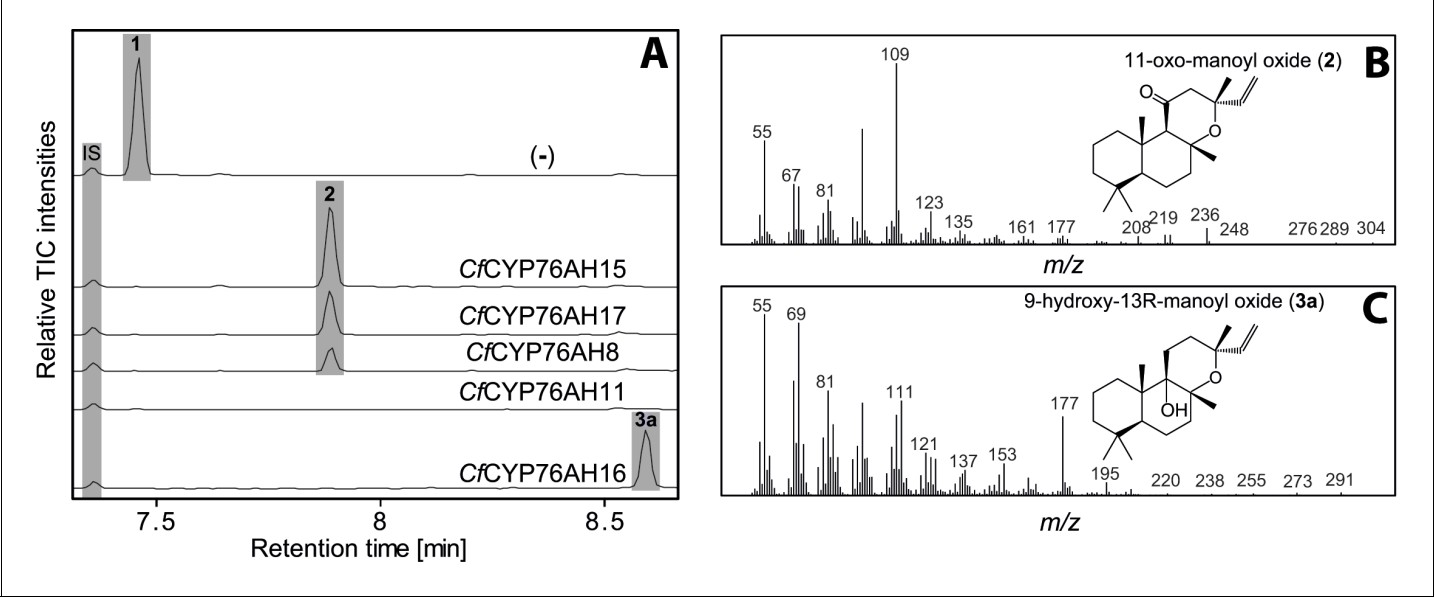

**Figure 3.** GC-MS analysis of 13*R*-manoyl oxide (1) derived diterpenoids obtained by transient expression of CYP76AHs from *C. forskohlii* in *N. benthamiana* leaves. (**A**) GC-MS total ion chromatograms (TIC) of extracts from *N. benthamiana* transiently expressing *CfCXS*, *CfGGPPS*, *CfTPS2* and *CfTPS3* (13*R*-manoyl oxide biosynthesis) genes in combination with water (-), *CfCYP76AH15*, *CfCYP76AH17*, *CfCYP76AH8*, *CfCYP76AH11* or *CfCYP76AH16*. 1-eicosene was used as internal standard (IS). 13*R*-manoyl oxide (1) was identified only in (-), indicating that it is further metabolized by the *CfCYP76AH15*, *CfCYP76AH17*, *CfCYP76AH8*, *CfCYP76AH11* and *CfCYP76AH16* enzymes. (**B**) *m/z* spectrum of 11-oxo-13*R*-manoyl oxide (2). (**C**) *m/z* spectrum of 9-hydroxy-13*R*-manoyl oxide (3a). The structure of both compounds was verified by NMR analysis (*Tables 1* and *2*). The compounds have been identified previously in *C. forskohlii* as putative intermediates in the *in planta* biosynthesis of forskolin (*Asada et al., 2012*). For each combination, extracts from leaves of three different *N. benthamiana* plants have been analyzed and representative chromatograms are shown.

The following figure supplements are available for figure 3:

**Figure supplement 1.** LC-qTOF-MS analysis of 13*R*-manoyl oxide-derived diterpenoids obtained by transient expression of *C. forskohlii* CYP76AH encoding genes in *N. benthamiana* leaves.

**Figure supplement 2.** GC-MS analysis of 13*R*-manoyl oxide-derived diterpenoids following transient expression in *N. benthamiana* leaves of the *C. forskohlii* gene encoding *CfCYP71D281* together with genes encoding the required enzymes for biosynthesis of 13*R*-manoyl oxide (*CfCXS*, *CfGGPPS*, *CfTPS2*, *CfTPS3*).

conversion of 13*R*-manoyl oxide to **2** with no concomitant formation of multi-oxygenated products (*Figure 2*). Compound **5d** produced by *CfCYP76AH8* as well as by *CfCYP76AH17* was identified as 1,11-dihydroxy-13*R*-manoyl oxide (*Figures 2B* and *4* and *Tables 1* and *2*). Noticable, this specific oxygenation pattern is found in forskolin. Minor amounts of several di- and trihydroxylated 13*R*-manoyl oxide-derived compounds were also produced by *CfCYP76AH8* and *CfCYP76AH17* (*Figure 2A* and *Figure 3—figure supplement 1*). Although we managed to identify the chemical structures of a number of 13*R*-manoyl oxide derivatives (*Figure 2B*), it was not possible to do so for all the compounds shown in *Figure 3—figure supplement 1*. The main obstacle was the high complexity as well as the small amounts of the diterpenoids produced in *N. benthamiana* leaves expressing the *CfCYP76AHs*. Additional limiting factors were the instability of several of these compounds and the limiting plant material available. Production of higher amounts of these compounds in microbial hosts was not pursued, because the terpenoid profiles observed following expression of the enzymes in plants and yeast cells were not identical.

Two additional CYPs of the CYP76AH subfamily catalyzed oxygenation of 13*R*-manoyl oxide at different positions without substantial formation of C-11 keto derivatives. *CfCYP76AH16* yielded predominantly **3a** which was identified by NMR as 9-hydroxy-13*R*-manoyl oxide (or coleorol) and *CfCYP76AH11* produced a range of monooxygenated derivatives including traces of **10b** which was identified by NMR as 9-deoxydeacetylforskolin (*Figure 2*). The positions of the carbonyl and

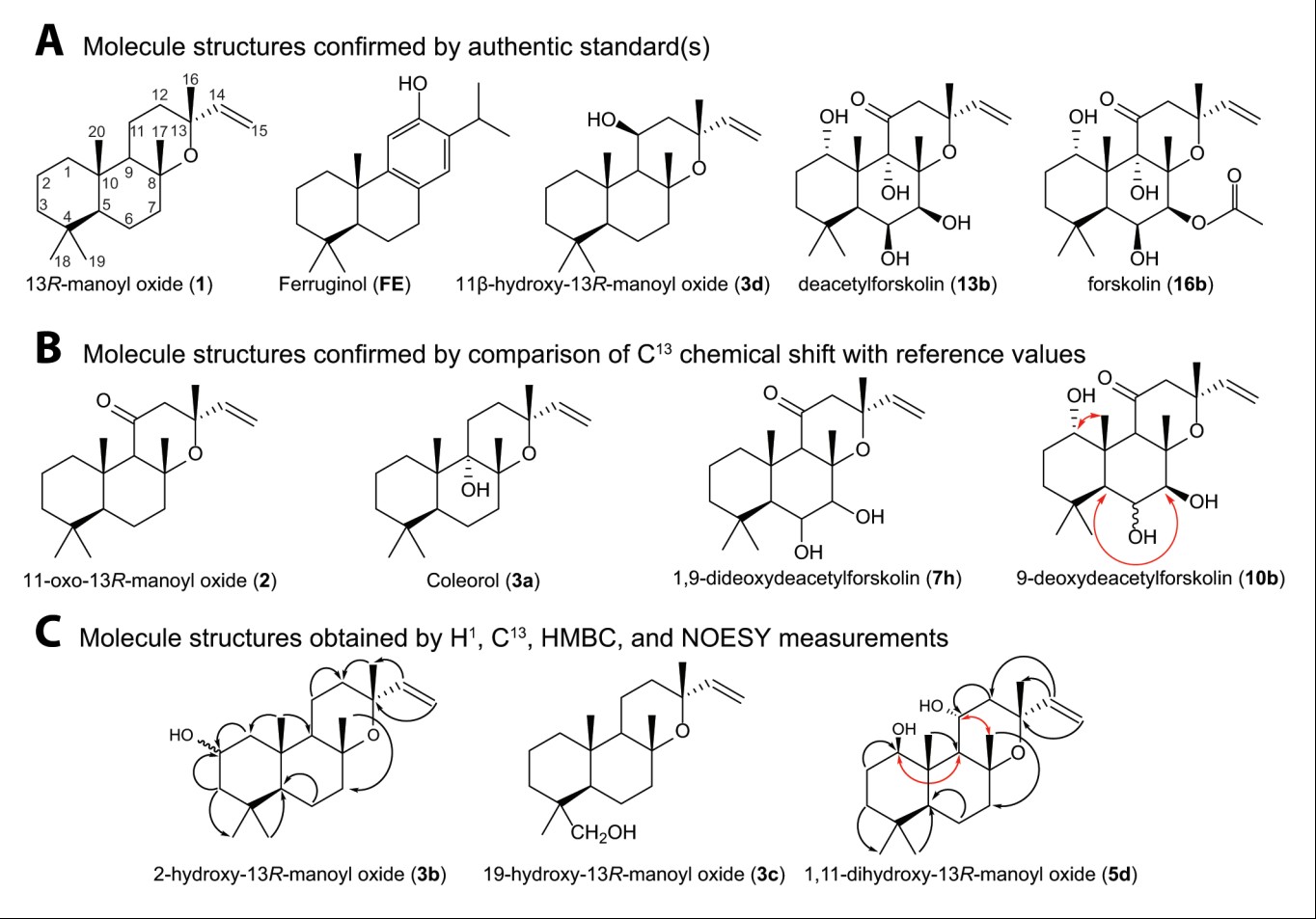

**Figure 4.** Structures of key compounds presented in this work. (A) Compounds confirmed using authentic standards. (B) Compounds which structure was confirmed/identified by comparison of $^{13}$C NMR data with existing literature. (C) Compounds which structure was confirmed/identified by HMBC and NOE correlations for assigning position of OH-groups (marked in red), whereas couplings identified in the previously uncharacterized compounds **3b**, **3c** and **5d** are marked in black. All other molecular structures were confirmed by $^{13}$C chemical shifts in comparisons to reference values (**Table 1**, **Figure 4—source data 1**).

The following source data is available for figure 4:

**Source data 1.** NMR spectra's of selected 13R-manoyl oxide derived molecules.

hydroxyl-groups in **3a** and **10b** were consistent with those in forskolin. Thus, the individual activities of the CYPs catalyzing formation of these compounds can be considered complementary in forskolin biosynthesis: **10b** carries the carbonyl-function at C-11 and the three hydroxyl groups observed in forskolin at positions C-1, C-6 and C-7, but lacks hydroxylation at the C-9 position, which is observed in **3a**. The only enzyme outside the CYP76AH subfamily that displayed activity toward 13R-manoyl oxide was CYP71D381, which resulted in oxidized derivatives at positions not compatible with forskolin (**Figure 3—figure supplement 2**).

To probe the role of the different *Cf*CYP76AH enzymes in forskolin biosynthesis, they were co-expressed by combining one of the three P450s catalyzing formation of **2** with the functionally distinct *Cf*CYP76AH11 and *Cf*CYP76AH16, first in pairs, then in all possible permutations as triplets (**Figure 5** and **Figure 5—figure supplement 1**). The combination of *Cf*CYP76AH15 and *Cf*CYP76AH11 afforded production of 6,7-dihydroxy-11-keto-manoyl oxide (**7h**) as part of a complex mixture. Formation of **7h** demonstrated combined introduction of the carbonyl group at C-11 together

**Table 1.** $^1$H-NMR and $^{13}$C-NMR chemical shifts (**Figure 4—source data 1**) of novel oxygenated 13R-(+)-manoyl oxide-derived diterpenoids formed following transient expression of CYP encoding genes from *C. forskohlii*.

| Pos. | 19-hydroxy-13R-manoyl oxide (3c)* $^1$H (nH; m; J(Hz)) | $^{13}$C | 2-hydroxy-13R-manoyl oxide (3b)* $^1$H (nH; m; J(Hz)) | $^{13}$C | 1,11-dihydroxy-13R-manoyl oxide (5d)* $^1$H (nH; m; J(Hz)) | $^{13}$C |
|---|---|---|---|---|---|---|
| 1 | 0.89 (1H;m)<br>1.63 (1H; m) | 39.1 | 1.10 (1H; t(br); 11.9, 11.9)<br>1.77 (1H; m) | 51.3 | 3.49 (1H; dd;11.1, 4.5) | 79.0 |
| 2 | 1.44 (1H; m)<br>1.56 (1H; m) | 18.1 | 3.92 (1H; m) | 65.3 | 1.75 (1H; td; 13.5, 11.1, 3.9)<br>1.60 (1H; m) | 29.0 |
| 3 | 0.95 (1H; m)<br>1.78 (1H; m) | 35.8 | 0.76 (1H; t(br); 11.9, 11.9)<br>1.99 (1H; d(br); 11.9) | 48.2 | 1.47 (1H; dd; 13.6, 3.9)<br>1.39 (1H; td; 13.5, 3.6) | 39.6 |
| 4 | | 38.5 | | 34.9 | | 33.4 |
| 5 | 1.10 (1H; dd; 2.3, 12.6) | 56.9 | 0.95 (1H; dd; 2.2, 12.4) | 55.9 | 0.84 (1H; dd; 11.3, 2.0) | 55.6 |
| 6 | 1.36 (1H; dd; 3.6, 12.6)<br>1.75 (1H; m) | 20.1 | 1.68 (1H; m)<br>1.27 (1H; m) | 19.7 | 1.47 (1H; m)<br>1.64 (1H; m) | 20.2 |
| 7 | 1.42 (1H; m)<br>1.83 (1H; dt; 3.3, 12.2) | 43.6 | 1.45 (1H; dd(br); 3.6, 12.5)<br>1.85 (1H; dt(br); 2.9, 12.5) | 43.2 | 1.48 (1H; m)<br>1.85 (1H; m) | 44.0 |
| 8 | | 75.1 | | 75.1 | | 75.3 |
| 9 | 1.35 (1H; dd; 4.3, 12.0) | 55.7 | 1.40 (1H; dd; 4.2, 11.9) | 55.4 | 1.54 (1H; d; 5.8) | 55.8 |
| 10 | | 37.3 | | 38.7 | | 43.8 |
| 11 | 1.48 (1H; m)<br>1.58 (1H; m) | 15.4 | 1.53 (1H; m)<br>1.61 (1H; m) | 15.6 | 4.38 (1H; br q; ≈8.6) | 65.6 |
| 12 | 1.78 (1H; m)<br>1.64 (1H; m) | 35.7 | 1.78 (1H; m)<br>1.66 (1H; m) | 35.5 | 2.02 (1H; dd; 14.3, 8.7)<br>2.27 (1H; dd; 14.3, 8.7) | 35.8 |
| 13 | | 73.4 | | 73.4 | | 72.8 |
| 14 | 5.87 (1H; dd; 10.8, 17.4) | 147.7 | 5.87 (1H; dd; 10.8, 17.4) | 147.7 | 5.90 (1H; dd; 17.4, 10.8) | 147.1 |
| 15 | 4.92 (1H; dd; 1.5, 10.8)<br>5.14 (1H; dd; 1.5, 17.4) | 110.2 | 4.92 (1H; d; 10.8)<br>5.14 (1H; d; 17.4) | 110.3 | 4.94 (1H; dd; 10.7, 1.5)<br>5.17 (1H; dd; 17.4, 1.5) | 111.2 |
| 16 | 1.27 (3H; s) | 28.5 | 1.27 (3H; s) | 28.7 | 1.27 (3H; s) | 32.1 |
| 17 | 1.28 (3H; s) | 25.3 | 1.29 (3H; s) | 25.7 | 1.49 (3H; s) | 27.8 |
| 18 | 0.97 (3H; s) | 26.8 | 0.93 (3H; s) | 33.5 | 0.78 (3H; s) | 13.5 |
| 19 | 3.70 (1H; d; 10.9)<br>3.46 (1H; d; 10.9) | 65.4 | 0.85 (3H; s) | 22.2 | 0.85 (3H; s) | 32.8 |
| 20 | 0.78 (3H; s) | 15.7 | 0.84 (3H; s) | 16.5 | 0.79 (3H; s) | 21.1 |

\* $^1$H and $^{13}$C NMR data acquired at 600 and 150 MHz, respectively, in methanol-$d_4$, at 300 K. s = singlet, d = doublet, t = triplet, m = multiplet, br = broad

with two hydroxyl groups at positions C-6 and C-7, again consistent with the oxygenation pattern of forskolin (**Figure 2** and **Figure 5—figure supplement 1**).

When the *Cf*CYP76AH enzymes were assayed in triplet combinations, the product profiles were further shifted towards multi-oxygenated 13R-manoyl oxide derivatives. The formation of minor amounts of deacetylforskolin (**13b**) and several compounds with identical mass to charge ratio (*m/z*) but different retention times were detected using different enzyme combinations (**Figure 5**, **Supplementary file 1**). The triplet combination *Cf*CYP76AH15, *Cf*CYP76AH11 and *Cf*CYP76AH16 led to the highest amounts of **13b**. Thus, this combination of multifunctional P450s appeared to constitute the optimal biosynthetic pathway for specific formation of **13b** from **1** (**Figure 5**).

**Table 2.** Structural identification of four oxygenated 13R-manoyl oxide-derived diterpenoids formed following transient expression of CYP encoding genes from C. forskohlii based on comparison of their ¹H-NMR and ¹³C-NMR (**Figure 4—source data 1**) chemical shifts to literature data. Chemical shifts for reference compounds marked with * have not been assigned to a specific carbon. The ¹³C chemical shifts of 9-deoxyforskolin (**Gabetta et al., 1989**) were used as reference for 6,7-dihydroxy-11-oxo-13R-manoyl oxide (7h).

| | 9-Deoxydeacetylforskolin (10b)[†] | | | 1,9-Dideoxydeacetylforskolin (7h)[†] | | | 11-oxo-13R-manoyl oxide (2)[†] | | Coleorol (3a)[†] | |
|---|---|---|---|---|---|---|---|---|---|---|
| Pos. | ¹H (nH; m; J(Hz)) | ¹³C | (Gabetta et al., 1989) | ¹H (nH; m; J(Hz)) | ¹³C | (Gabetta et al., 1989) | ¹³C | (Gabetta et al., 1989) | ¹³C | (Asada et al., 2012) |
| 1 | 4.38 (1H; t; 2.8) | 71.6 | 71.2 | 2.45 (1H, d(br); 13.1) 0.78 (H; m) | 41.5 | 43.1 | 42.1 | 41.9 | 31.7 | 31.6 |
| 2 | 1.47 (1H; m) 2.14 (1H; m) | 25.8 | 25.6 | 1.78 (H; m) 1.40 (H; m) | 18.7 | 18.4 | 18.5 | 18.4 | 18.6 | 18.4 |
| 3 | 1.12 (1H; dt; 3.4, 13.2) 1.62 (1H; dt; 3.5, 13.5) | 36.4 | 36.3 | 1.36 (H; m) 1.15 (H; m) | 43.8 | 43.7 | 43.4 | 43.3 | 41.9 | 41.8 |
| 4 | | 34.2 | 34.1 | | 34.4 | 34.1 | 33.4 | 33.2 | 33.3 | 33.2 |
| 5 | 1.34 (1H; d; 2.1) | 47.5 | 47.4 | n.d. | 55.7 | 55.2 | 56.0 | 55.8 | 45.7 | 45.5 |
| 6 | 4.44 (1H; t; 2.6) | 70.8 | 70.2 | 4.39 (1H; m) | 70.4 | 70.2 | 19.8 | 19.7 | 19.5 | 19.4 |
| 7 | 3.68 (1H; d; 3.6) | 80.7 | 81.1 | 3.71 (1H; d; 3.8) | 81.0 | 80.7 | 39.6 | 39.4 | 36.6 | 36.4 |
| 8 | | 80.0 | 78.5 | | 80.1 | 79.9 | 77.5 | 77.2 | 78.0 | 77.8 |
| 9 | 3.32 (1H; s) | 58.0 | 58.2 | 2.59 (1H; s) | 65.5 | 65.4 | 66.9 | 66.7 | 75.3 | 75.2 |
| 10 | | 42.2 | 41.7 | | 38.0 | 37.8 | 37.3 | 37.1 | 41.1 | 40.9 |
| 11 | | 207.7 | 207.6 | | 206.3 | 205.7 | 207.7 | 207.1 | 21.1 | 21.0 |
| 12 | 2.63 (1H; d; 18.0) 2.69 (1H; d; 18.0) | 49.8 | 49.9 | 2.60 (1H; d; 18.1) 2.66 (1H; d; 18.1) | 50.0 | 49.8 | 50.4 | 50.2 | 31.6 | 31.5 |
| 13 | | 75.1 | 74.8 | | 75.1 | 75.1 | 75.1 | 74.4 | 72.9 | 72.8 |
| 14 | 5.94 (1H; dd; 10.8, 17.4) | 146.2 | 145.8 | 5.95 (1H; dd; 10.7, 17.4) | 146.9 | 146.4 | 146.9 | 146.0 | 147.4 | 147.3 |
| 15 | 5.04 (1H; d; 10.8) 5.14 (1H; d; 17.4) | 112.4 | 112.7 | 5.04 (1H; d; 10.7) 5.17 (1H; d; 17.4) | 112.3 | 112.1 | 112.3 | 111.9 | 110.1 | 110.0 |
| 16 | 1.30 (3H; s) | 31.5 | 31.5* | 1.28 (3H; s) | 31.6 | 33.2* | 31.4 | 31.2* | 28.9 | 28.8 |
| 17 | 1.54 (3H; s) | 24.1 | 24.5* | 1.50 (3H; s) | 23.5 | 31.4* | 28.1 | 27.9* | 27.0 | 29.9 |
| 18 | 1.38 (3H; s) | 33.1 | 18.2* | 0.97 (3H; s) | 33.4 | 23.9* | 15.6 | 15.5* | 33.7 | 33.6 |
| 19 | 1.21 (3H; s) | 23.7 | 23.6* | 1.21 (3H; s) | 24.0 | 23.7* | 21.8 | 21.6* | 21.5 | 21.4 |
| 20 | 1.01 (3H; s) | 18.5 | 32.8* | 1.30 (3H; s) | 17.2 | 16.7* | 33.6 | 33.5* | 17.0 | 16.8 |

[†]¹H and ¹³C NMR data acquired at 600 and 150 MHz, respectively, in methanol-$d_4$, at 300 K. s = singlet, d = doublet, t = triplet, m = multiplet, br = broad

## Monooxygenase activity of the CYP76AH subfamily toward other terpene scaffolds

In addition to 13R-manoyl oxide-derived diterpenoids, the root cork of C. forskohlii contains numerous abietane diterpenoids derived from miltiradiene (**Alasbahi and Melzig, 2010a**). Recently a pair of diterpene synthases (CfTPS1 and CfTPS3) mainly expressed in the root cork of C. forskohlii was demonstrated to produce miltiradiene (**Pateraki et al., 2014**). It has been shown previously that members of the CYP76AH subfamily in Lamiaceae are able to oxygenate miltiradiene or derivatives thereof. CYP76AH1 from Salvia miltiorrhiza (**Guo et al., 2013**), CYP76AH4 as well as RoFS1 and RoFS2 from Rosmarinus officinalis (**Božić et al., 2015**; **Zi and Peters, 2013**) and CYP76AH24 from S. pomifera catalyze synthesis of ferruginol, the precursor of carnosic acid and tanshinones (**Guo et al., 2013**; **Ignea et al., 2016a**), from miltiradiene or dehydroabietadiene. Additionally, CYP76AH3 from

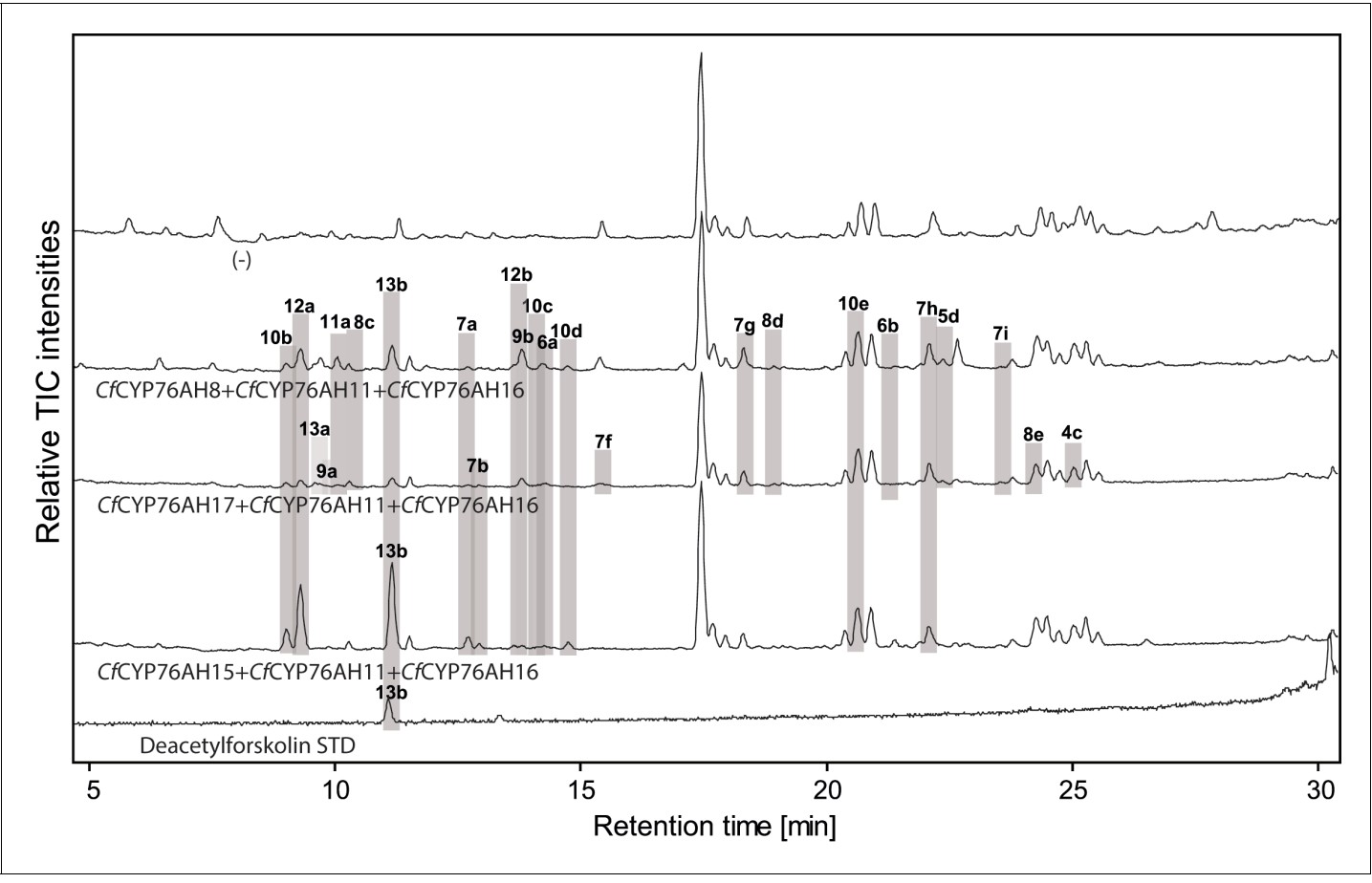

**Figure 5.** LC-qTOF-MS analysis of 13*R*-manoyl oxide-derived diterpenoids obtained by transient expression of combinations of *C. forskohlii* CYP encoding genes, together with genes encoding the required enzymes for biosynthesis of 13*R*-manoyl oxide in *N. benthamiana* leaves. Total ion chromatograms (TIC) of extracts expressing the 13*R*-manoyl oxide biosynthesis genes (*CfCXS*, *CfGGPPS*, *CfTPS2*, *CfTPS3*), in combination with (from the top) water (-), *CfCYP76AH8 + CfCYP76AH11 + CfCYP76AH16*, *CfCYP76AH17 + CfCYP76AH11 + CfCYP76AH16*, and *CfCYP76AH15 + CfCYP76AH11 + CfCYP76AH16* are shown. Hydroxylated 13*R*-manoyl oxide-derived diterpenoids (marked with grey bars) and their identity including their molecular formulas were confirmed by accurate mass (5 ppm tolerance, *Supplementary file 1*). Compounds present in trace amounts are not marked. The identity of 1,11-dihydroxy-13*R*-manoyl oxide (**5d**), 9-deoxydeacetylforskolin (**10b**) and 1,9-dideoxydeacetylforskolin (**7h**) was confirmed by NMR analysis (*Figure 4* and *Tables 1* and *2*), whereas the identity of deacetylforskolin (**13b**) was confirmed by comparison to an authentic chemically synthesized standard. No 13*R*-manoyl oxide-derived diterpenoids were identified in the water control (-). For each combinaton, extracts from leaves of three different *N. benthamiana* plants have been analyzed and representative chromatograms are shown.

The following figure supplement is available for figure 5:

**Figure supplement 1.** LC-qTOF-MS analysis of 13*R*-manoyl oxide-derived diterpenoids obtained by transient expression of combinations of *C. forskohlii* CYP76AH encoding genes in *N. benthamiana* leaves.

*S. miltiorrhiza* has been shown to accept ferruginol as a substrate to produce sugiol, 11-hydroxy-ferruginol and 11-hydroxy-sugiol (*Guo et al., 2016*). These miltiradiene-accepting CYP76AHs show high-sequence homology, ranging from 60% to 85% at the amino acid level, with those identified in *C. forskohlii*. Therefore, it was tempting to study the ability of the *Cf*CYP76AHs to metabolize miltiradiene. The *Cf*CYP76AHs were co-expressed individually in *N. benthamiana* leaves producing miltiradiene (*Andersen-Ranberg et al., 2016*), and the product profile monitored by unbiased LC-MS analysis. *Cf*CYP76AH15 was shown to convert miltiradiene to ferruginol (*Figure 6*). Ferruginol was identified in extracts of *C. forskohlii* root cork (*Figure 6*), so it is possible that *Cf*CYP76AH15 is also involved in the biosynthesis of ferruginol *in planta*. In a parallel series of experiments, CYP76AHs from rosemary and salvia known to accept miltiradiene as substrate were tested for their ability to

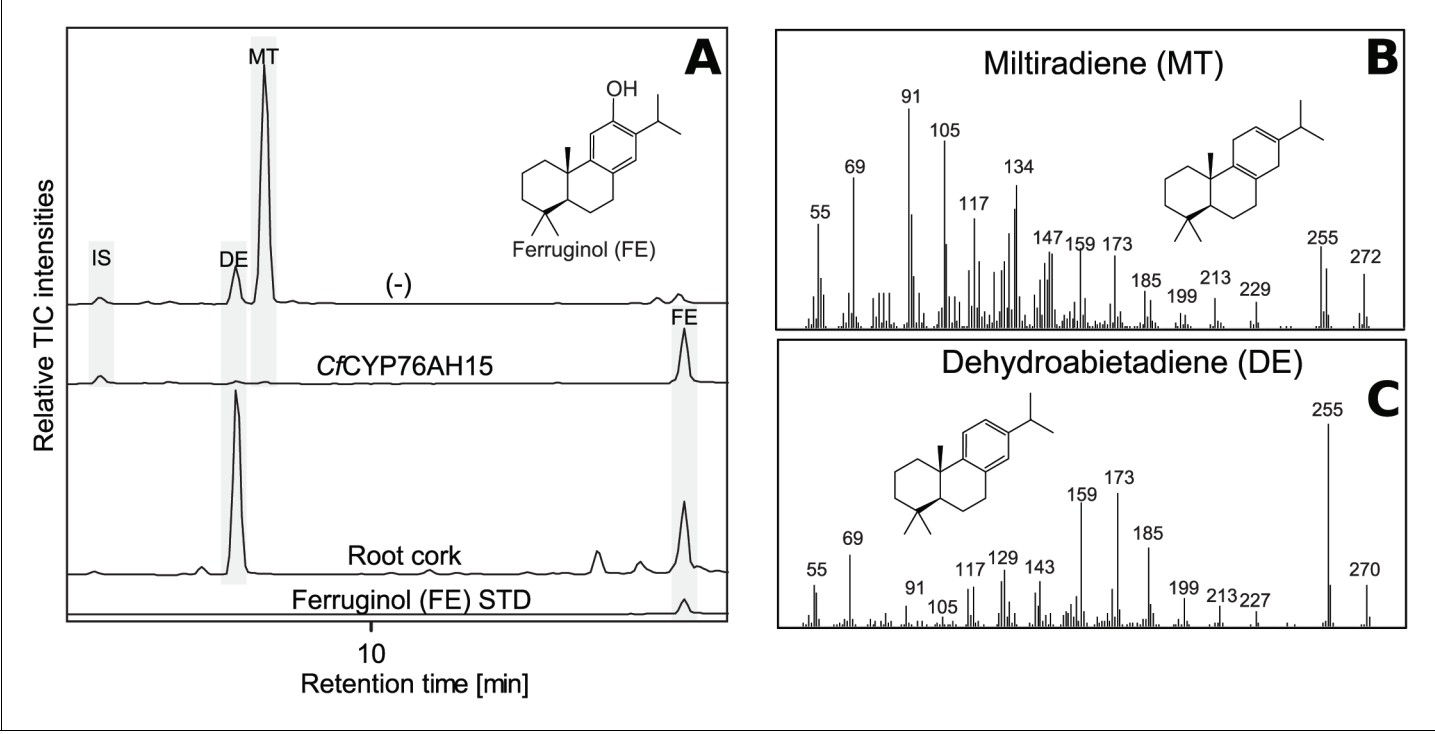

**Figure 6.** GC-MS analysis of miltiradiene-derived diterpenoids obtained by transient expression of *CfCYP76AH15* in *N. benthamiana* leaves. (A) Total ion chromatograms (TIC) of extracts transiently expressing *CfCXS, CfGGPPS, CfTPS1* and *CfTPS3* (miltiradiene biosynthesis genes) in combination with water (-) or *CfCYP76AH15*. Dehydroabietadiene (DE) and miltiradiene (MT) were observed in the (-) extract, whereas ferruginol was observed in extracts from tissue expressing the miltiradiene biosynthesis genes together with *CfCYP76AH15*. In root cork extracts, ferruginol was detected together with dehydroabietadiene. Presence of ferruginol was confirmed by comparison to an authentic standard (*Ignea et al., 2016a*), while presence of miltiradiene (B) and dehydroabietadiene (C) were confirmed by comparison of *m/z* spectra with previously characterized compounds (*Andersen-Ranberg et al., 2016*). For every combination, extracts from leaves of three different *N. benthamiana* plants have been analyzed and representative chromatograms are shown.

use 13*R*-manoyl oxide as a substrate. Transient expression of *Ro*CYP76AH4 (*Zi and Peters, 2013*), *Ro*FS1 and *Sf*FS (*Božić et al., 2015*) in *N. benthamiana* leaves able to synthesize 13*R*-manoyl oxide demonstrated that *Ro*CYP76AH4 was able to efficiently convert 13*R*-manoyl oxide to 11-oxo-13*R*-manoyl, while *Ro*FS1 and *Sf*FS were able to produce 11-hydroxy manoyl oxide, in addition to 11-oxo-13*R*-manoyl oxide (*Figure 7*).

## Establishing regiospecific acetylation as the final step of forskolin biosynthesis

To complete the biosynthetic route to forskolin, specific acetylation of the C-7 hydroxyl group of deacetylforskolin (**13b**) is required. The root cork transcriptome was mined for acyltransferases (ACTs) from clade III of the BAHD family earlier reported to predominantly use acetyl CoA as acetyl donor for acetylation of hydroxyl groups (*D'Auria, 2006*). Ten ACTs (*Figure 1—source data 1*) were identified, cloned and tested functionally by *Agrobacterium*-mediated transient expression in *N. benthamiana* leaves, engineered to produce deacetylforskolin by co-expression of the enzymes *Cf*DXS, *Cf*GGPPS, *Cf*TPS2, *Cf*TPS3, *Cf*CYP76AH15, *Cf*CYP76AH11 and *Cf*CYP76AH16. Two ACT candidates, *Cf*ACT1-6 and *Cf*ACT1-8, were found to catalyze acetylation of **13b** (*Figure 8*). Expression of *Cf*ACT1-6 resulted in formation of a broad range of acetylated products of which forskolin constituted a minor fraction. In contrast, *Cf*ACT1-8 exhibited high activity and specificity, with efficient conversion of **13b** to forskolin and absence of detectable acetylated side products. Identification of this enzyme establishes the entire and highly specific biosynthetic route to forskolin from its precursor, GGPP (*Figure 9*).

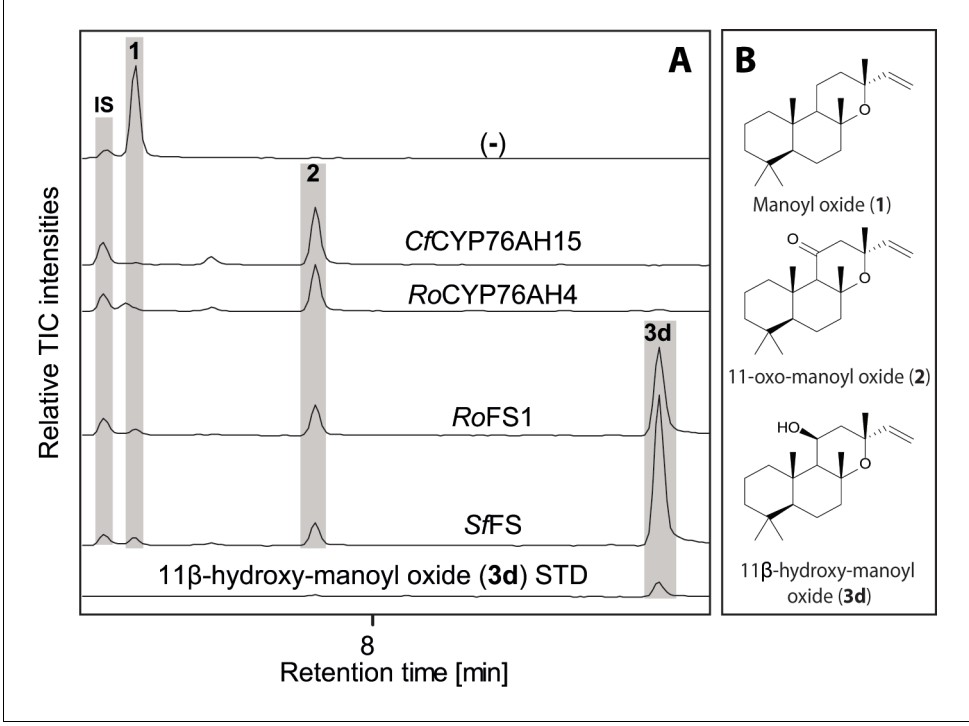

**Figure 7.** GC-MS analysis of 13*R*-manoyl oxide-derived diterpenoids obtained by transient expression of *CYP76AHs* in *N. benthamiana* leaves. (**A**) Total ion chromatograms (TIC) of extracts transiently expressing *CfCXS*, *CfGGPPS*, *CfTPS2* and *CfTPS3* (13*R*-manoyl oxide biosynthesis genes) in combination with water (-), *CfCYP76AH15*, *RoCYP76AH4*, *RoFS1* and *SpFS* are shown. 13*R*-manoyl oxide was observed in the (-) extracts, while 11-oxo-13*R*-manoyl oxide (**2**) was observed in the *CfCYP76AH15*, *RoCYP76AH4*, *RoFS1* and *SfFS* extracts. 11-Hydroxy-13*R*-manoyl oxide (**3d**) is observed only in extracts expressing the *RoFS1* and *SfFS1* genes. Presence of 11-hydroxy-13*R*-manoyl oxide was verified by comparison to an authentic standard (**Ignea et al., 2016b**) while identification of 11-oxo-13*R*-manoyl oxide was confirmed by comparison of *m/z* spectra with a previously characterized compound (**2**). The results show *RoCYP76AH4* has an activity similar to *CfCYP76AH15*, able to convert efficiently and specifically 13*R*-manoyl oxide to **2**. *RoFS1*, as well as *SfFS*, can also convert 13*R*-manoyl oxide to **2** but they catalyze the synthesis of an additional product, 11-hydroxy-13*R*-manoyl oxide (**3d**). For every combination, extracts from leaves of three different *N. benthamiana* plants have been analyzed and representative chromatograms are shown.

## Engineering of the entire pathway of forskolin in *Saccharomyces cerevisiae*

Expression and engineering of plant biosynthetic pathways in microbial organisms provides a method for sustainable production of high value compounds like the structurally complex bioactive diterpenoids (*Guo et al., 2013*; *Ignea et al., 2016a*; *Jia et al., 2016*). With the genes encoding the entire biosynthetic pathway of forskolin identified, we proceeded to establish stable forskolin production in *S. cerevisiae*, an excellent host organism for biosustainable and scalable production of numerous bio-active natural products (*Brochado et al., 2010*; *Brown et al., 2015*; *Galanie et al., 2015*; *Hansen et al., 2009*; *Ignea et al., 2016a*; *Jeandet et al., 2012*). For expression in yeast, all *C. forskohlii* genes were codon-optimized and stably integrated in neutral loci in the yeast genome. Genomic integration was chosen versus expression via episomal plasmids as the former strategy favors the simultaneous expression of a large number of genes as well as effective selection marker recycling (*Jensen et al., 2014*). Additionally, a sequence encoding a NADPH-dependent cytochrome P450 oxidoreductase (*CfPOR*), required to support the P450 activity, was identified from the *C. forskohlii* root cork transcriptome and cloned for co-integration in the yeast genome. The isolated *CfPOR* was the only one present in the root cork transcriptome. Genomic integration enabled stable, simultaneous expression of a total of eight heterologous genes in the microbial host.

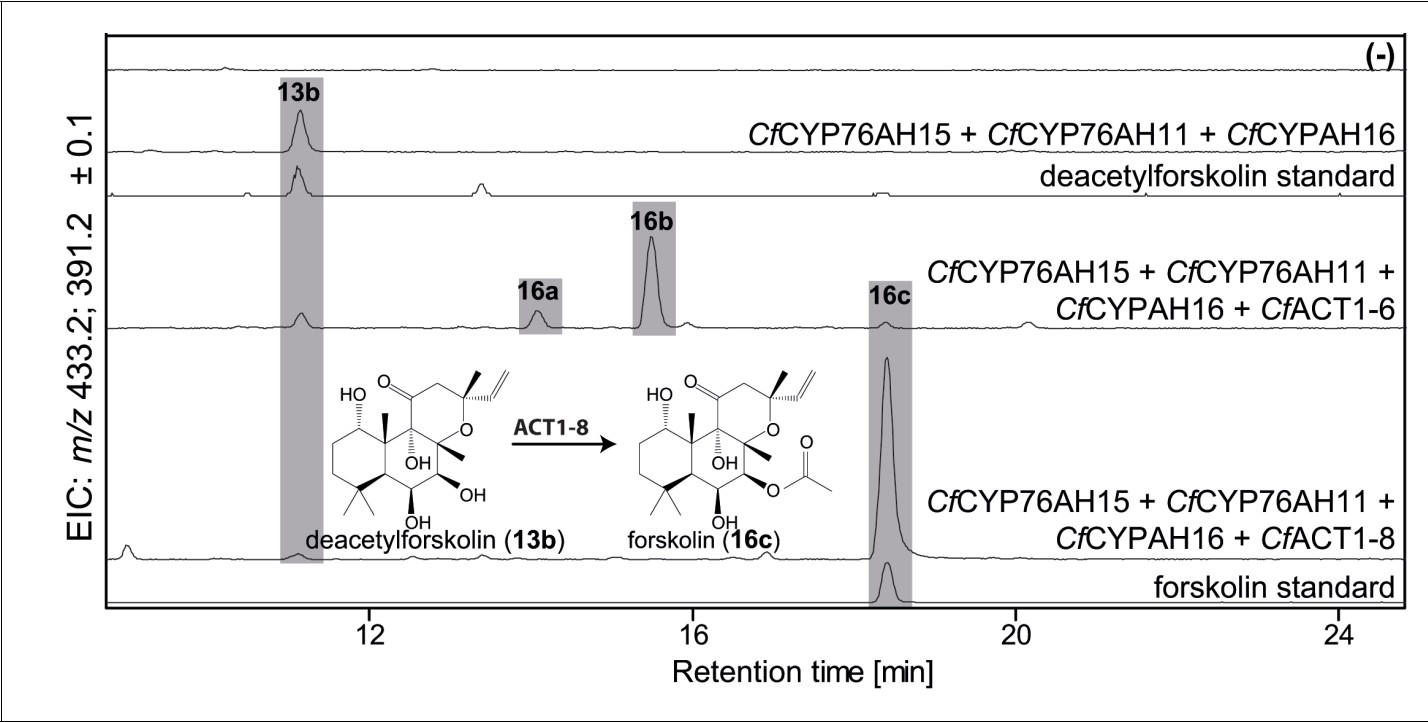

**Figure 8.** De novo biosynthesis of forskolin by transient expression of *C. forskohlii* genes in *N. benthamiana* as monitored by LC-MS-based extracted ion chromatograms (EIC). To monitor both deacetylforskolin (**13b**) and forskolin (**16c**), the EIC were selected as the sum of *m/z* 433.2 ± 0.1 and m/z 391.2 ± 0.1. Chromatograms represent LC-MS analysis of extracts from leaves expressing the 13*R*-manoyl oxide biosynthesis genes (*CfDXS, CfGGPPS, CfTP2* and *CfTPS3*) in combination with (from the top): water (-); *CfCYP76AH15, CfCYP76AH11* and *CfCYPAH16; CfCYP76AH15, CfCYP76AH11, CfCYPAH16* and *CfACT1-6; CfCYP76AH15, CfCYP76AH11, CfCYPAH16* and *CfACT1-8,* shown together with authentic standards (deacetylforskolin and forskolin). Forskolin (**16c**) was identified together with two other acetylated compounds (e.g. **16a, 16b**) with the same molecular mass in leaves expressing *CfACT1-6* together with forskolin-specific CYPs (***Supplementary file 1***). When *CfACT1-8* was expressed instead of *CfACT1-6*, a predominant accumulation of forskolin was observed, with a drastic reduction of non-specific acetylated products. For all combinations, extracts from leaves of three different *N. benthamiana* plants have been analyzed and representative chromatograms are shown.

*Cf*POR, *Cf*CYP76AH15, *Cf*CYP76AH11, *Cf*CYP76AH16 and *Cf*ACT1-8 were co-expressed in the yeast strain EFSC4498, previously engineered to produce 350 mg/L 13*R*-manoyl oxide (***Andersen-Ranberg et al., 2016***). Transformed yeast strains verified for the integration of all forskolin biosynthetic cDNAs into their genome, were further analyzed and production titers of forskolin and pathway intermediates were monitored. The highest forskolin producing strain, EVST21543, which demonstrated genetic stability through several rounds of cultivation, was grown in a 5 L fermenter using minimal medium under glucose-limited conditions. During fermentation, accumulation of forskolin (**16c**), 13*R*-manoyl oxide (**1**), 9-hydroxy-13*R*-manoyl oxide (**3a**), and biomass formation (***Figure 10***) were monitored over time. Forskolin levels reached more than 40 mg/L of yeast culture. Simultaneous accumulation of high titers of 13*R*-manoyl oxide (**1**) and 9-hydroxy-13*R*-manoyl oxide (**3a**) at levels of 200 and 500 mg/L, respectively, indicated that the conversion of 13*R*-manoyl oxide (**1**) to forskolin was far from complete. Interestingly, these intermediates did not accumulate in the *C. forskohlii* root cork. A comparison of the total intermediate profiles of *C. forskohlii* root cork versus the fermenter grown EVST21543 yeast strain is shown in (***Figure 10—figure supplement 1***).

## Discussion

The terpenoid biosynthetic pathways active in the root of *C. forskohlii* produce an array of 13*R*-manoyl oxide and miltiradiene-derived diterpenoids, including forskolin. Forskolin is one of the most complex and highly oxygenated diterpenoids reported in *C. forskohlii*. In the current study, the genes encoding the entire biosynthetic pathway for forskolin were identified. Availability of the

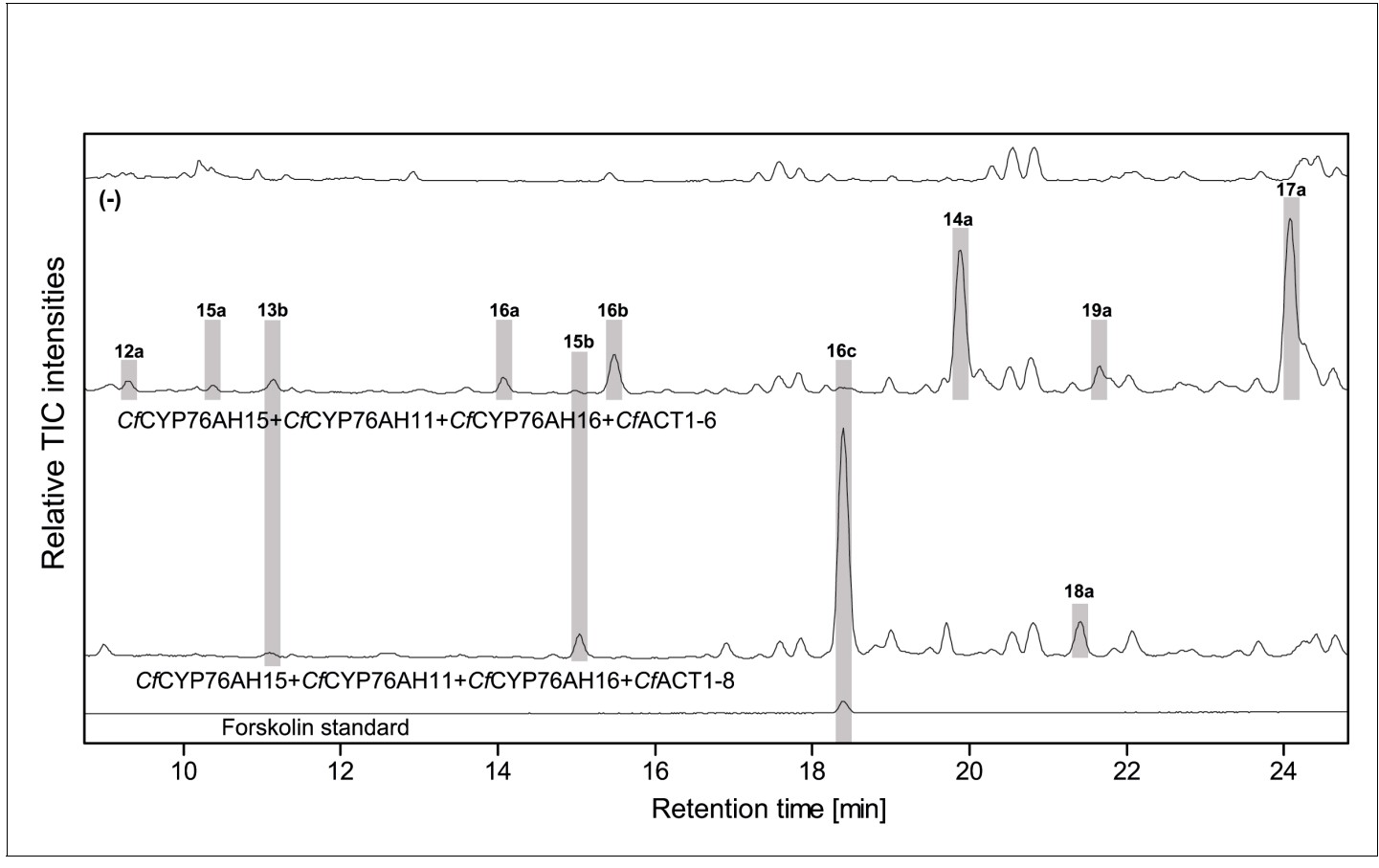

**Figure 9.** LC-qTOF-MS analysis of 13*R*-manoyl oxide-derived diterpenoids obtained by transient expression of combinations of *C.forskohlii* CYP and ACT encoding genes in *N. benthamiana* leaves. Total ion chromatograms (TIC) from extracts expressing the 13*R*-manoyl oxide biosynthesis genes (*CfCXS, CfGGPPS, CfTPS2, CfTPS3*) in combination with (from the top) water (-), *CfCYP76AH15 + CfCYP76AH11+ CfCYP76AH16 + CfACT1-6*, and *CfCYP76AH15 + CfCYP76AH11 + CfCYP76AH16 + CfACT1-8* are shown. Hydroxylated and acetylated 13*R*-manoyl oxide-derived diterpenoids (marked with grey bars) and their identity, including their molecular formulas, was confirmed by their accurate mass (5 ppm tolerance, **Supplementary file 1**). Compounds present in trace amounts are not marked. The identity of deacetylforskolin (**13b**) and forskolin (**16c**) was confirmed by comparison to authentic standards. No 13*R*-manoyl oxide-derived diterpenoids were detected in the water control (−). For all combinations, extracts from leaves of three different *N. benthamiana* plants have been analyzed and representative chromatograms are shown.

transcriptome from the root cork cells of *C. forskohlii,* where forskolin biosynthesis takes place, the option to achieve rapid functional characterization of the gene candidates *in planta* by transient expression in *N. benthamiana* and high-sensitivity techniques for structural characterization made the pathway elucidation possible. With the genes encoding the entire forskolin biosynthetic pathway in hand, de novo production of forskolin in engineered yeast was achieved.

Initially, a number of genes encoding CYP76AH subfamily members, expressed mainly in the root cork of *C. forskohlii*, was cloned and transiently expressed in *N. benthamiana* leaves being able to produce 13*R*-manoyl oxide. The products profiles obtained with these enzymes revealed that the identified *Cf*CYP76AHs have discrete roles in forskolin biosynthesis. Efficient monooxygenation at C-11 is catalyzed mainly by *Cf*CYP76AH15 (but also by *Cf*CYP76AH8, *Cf*CYP76AH17 and *Cf*CY-P76AH11). Monooxygenation at position C-9 is catalyzed exclusively by *Cf*CYP76AH16 and monooxygenation at C-1, C-6 and C-7 mainly by *Cf*CYP76AH11. Monooxygenation at C-1 was also observed using CfCYP76AH8 and CfCYP76AH17 (**4c**, *Figure 3—figure supplement 1*). Collectively, this set of data displays the multifunctional roles of these enzymes. Together, they could account for all the oxygenated positions in forskolin. Co-expression of *Cf*CYP76AH15, *Cf*CYP76AH11 and *Cf*CY-P76AH16 resulted in specific and efficient formation of the final intermediate, deacetylforskolin.

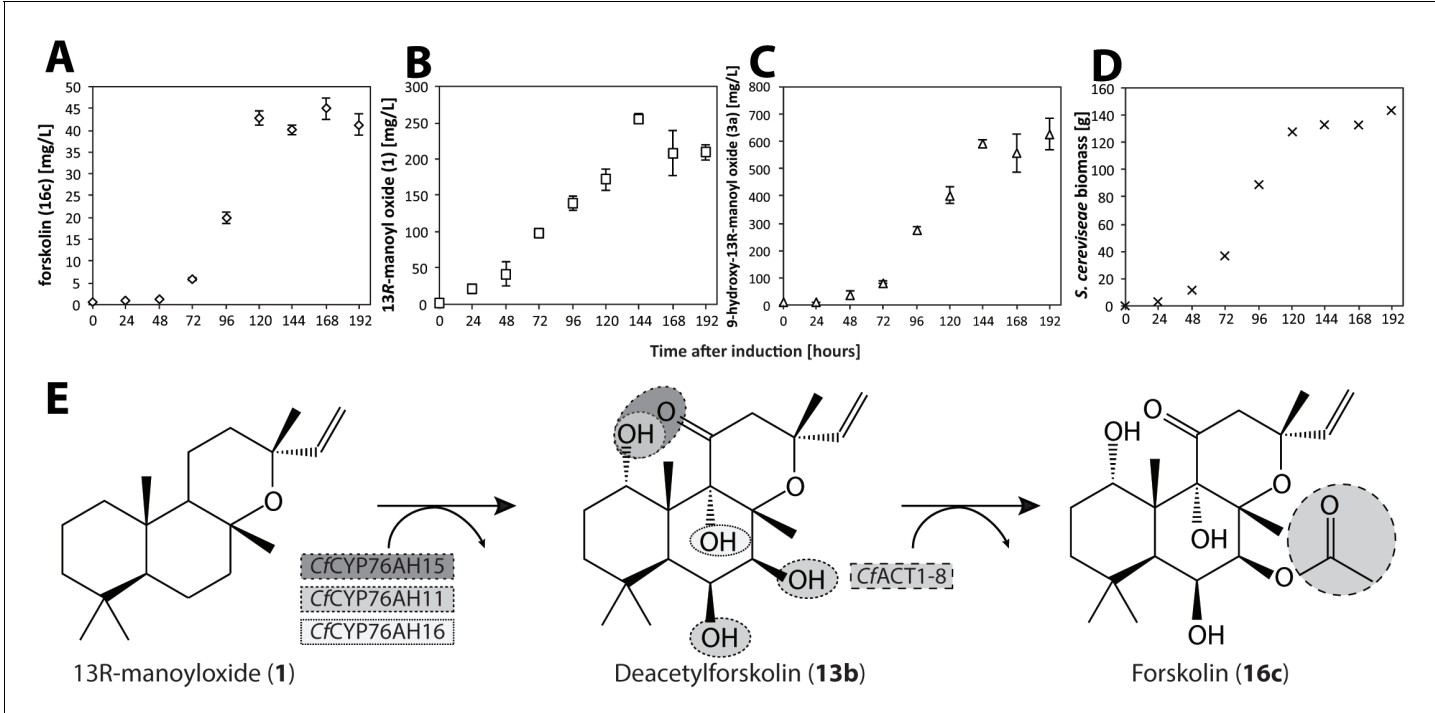

**Figure 10.** Forskolin production in *S. cerevisiae* following stable genomic integration of codon-optimized *C.forskohlii* genes. (A) Forskolin (**16c**) accumulation in a fermenter batch using the EVST21543 strain (expressing *Cf*CYP76AH15, *Cf*CYP76HA11, *Cf*CYP76AH16 and *Cf*ACT1-8 encoding genes in the EFSC4498 *S. cerevisiae* strain, optimized for the production of 13*R*-manoyl oxide [*Andersen-Ranberg et al., 2016*]). (B) 13*R*-manoyl oxide (**1**) accumulation in EVST21543 strain. (C) 9-Hydroxy-13*R*-manoyl oxide (**3a**) accumulation in EVST21543 strain. (D) EVST21543 strain biomass monitored during the fermentation process. (E) The biosynthetic pathway used for the production of forskolin in yeast. The fermentation event occurred once, and a triplicate of samples were analysed from each time course.

The following figure supplement is available for figure 10:

**Figure supplement 1.** Comparison of metabolite profiles between fermenter grown yeast culture of the EVST21543 strain and *C.forskohlii* root extract analyzed by LC-MS.

Despite the complementary multifunctionality of the CYP76AH enzymes, partial functional redundancy is possible, as demonstrated by the varied oxygenation patterns observed in experiments with single enzymes. Overlapping functionalities may contribute to a coordinated action for efficient conversion of **1** to **13b**. Moreover, certain *C. forskohlii* CYP76AH enzymes seem able to accept oxygenated forms of **1** as substrates which results in the observed shifts in profile toward higher decorated products when they are co-expressed in *N. benthamiana* (*Supplementary file 1*).

Although co-expression of *Cf*CYP76AH11 and *Cf*CYP76AH16 with either one of the three *Cf*CYP76AH8, *Cf*CYP76AH17 or *Cf*CYP76AH15 in the engineered system resulted in biosynthesis of deacetylforskolin (**13b**) (*Figure 5*), the precise sequence of *in planta* 13*R*-manoyl oxide oxygenations cannot be deduced from the experimental results partly because all identified *Cf*CYP76AHs accept **1** as substrate. The co-expression of the *Cf*CYP76AH encoding genes in the root cork of *C. forskohlii* and their partial functional redundancy or complementarity may in vivo constitute the basis for the chemical diversity of labdane terpenoids present in the root cork of *C. forskohlii*. *In planta*, the forskolin biosynthetic pathway would thus appear to be entangled within a metabolic grid offering simultaneous production of a multitude of other diterpenoids.

Recently, CYP76AH enzymes accepting miltiradiene (a non-epoxylabdane, abietane diterpenoid, which is also present in *C. forskohlii* roots) as substrate were reported from other Lamiaceae species (*Božić et al., 2015*; *Guo et al., 2016*; *Ignea et al., 2016a*; *Zi and Peters, 2013*). This prompted us to examine whether the promiscuous and multifunctional *Cf*CYP76AH could accept miltiradiene as substrate, and vice versa, e.g. if different Lamiaceae CYP76AHs can catalyze oxygenations of 13*R*-

manoyl oxide (*Figures 6* and *7*). According to our results *Cf*CYP76AH15 was found to have very similar catalytic activities compared to the rosemary CYP76AH4, an enzyme with a suggested role in the oxygenation of miltiradiene toward the synthesis of ferruginol (*Zi and Peters, 2013*). Efficient formation of ferruginol as well as 11-oxo-13*R*-manoyl oxide, by both enzymes indicates that they may represent orthologues. Two additional ferruginol synthases of the CYP76AH subfamily, one from *Salvia fruticosa* (*Sf*FS) and one from *Rosemary officinalis* (*Ro*FS), were found to catalyze the conversion of 13*R*-manoyl oxide to 11-oxo-13*R*-manoyl oxide and 11-hydroxy-manoyl oxide when expressed in *N. benthamiana* leaves (*Figures 6* and *7*). These findings are not reflected in the phylogenetic analysis of the known CYP76AHs. All *C. forskohlii* CYP76AHs able to produce 11-oxo-13*R*-manoyl oxide are clustered together, while CYP76AHs from *Salvia* spp. and *R. officinalis* that can catalyze the formation of the same compound, as well as those CYPs able to accept miltiradiene as substrate, form a different cluster when analyzed with currently known CYP76AHs. Thus, it is likely that the ability of CYP76AHs to catalyze 11-oxo-13*R*-manoyl oxide has evolved convergently in these plants (*Figure 11*).

These data highlight the functional versatility of the CYP76 family. The enzymes can exhibit broad substrate specificity which may advance metabolic evolution as they provide metabolic plasticity and flexibility affording synthesis of new diterpenoids. This facilitates the expansion of the number of possible diterpenoids produced in nature and potentially serves to diversify and augment the phytochemical defense of plants. The promiscuity of the CYP76AHs also provides potentials for their use in combinatorial approaches for synthesizing a range of diterpenoids with pharmaceutical relevance. The exclusive presence of the CYP76AH subfamily in Lamiaceae species may reflect gene duplications in a Lamiaceae ancestral species followed by expansion and neofunctionalization after speciation (*Figure 11*).

The identification of *Cf*ACT1-8 as an ACT catalyzing regiospecific acetylation of deacetylforskolin (**13b**) to afford forskolin completed the entire biosynthetic pathway for forskolin from its precursor, GGPP. Interestingly, the only currently identified acyltransferases in diterpenoid biosynthesis are those involved in the biosynthesis of paclitaxel. Those acetyltransferases show substantial regioselective promiscuity (*Ondari and Walker, 2008*; *Walker and Croteau, 2000*) and belong to Clade V (*D'Auria, 2006*; *Tuominen et al., 2011*), whereas the majority of the ACTs identified in the root cork transcriptome of *C. forskohlii*, including ACT1-6 and ACT1-8, belong to Clade III (*Figure 12*).

With all forskolin biosynthetic pathway genes identified, we moved to the generation of a stable forskolin producing *S. cerevisiae* strain. To engineer a stable microbial production platform, we proceeded to integrate the minimum required set of functional parts into the *S. cerevisiae* genome. Specific de novo production of the highly functionalized diterpenoid at titers above 40 mg/L was achieved through a pathway consisting of a total of 10 enzymatic steps catalyzed by eight heterologously co-expressed enzymes. This high forskolin titer, achieved with no optimization steps, highlights the potential to develop a microbial manufacturing platform for efficient and stereospecific production of forskolin with further fine-tuning of the biosynthetic pathway. Currently, it is not possible to accurately estimate the forskolin titers necessary for industrially profitable production, as the exact commercial applications, market size and price as well as the production cost including downstream processing cannot be determined. Given the knowledge gained in our present study and experiences with other compounds (*Paddon and Keasling, 2014*) we find it realistic to aim for yields ranging from a single to double digits of gram per liter of yeast culture. To achieve higher forskolin yields, it is important upfront to ascertain a proper flux toward GGPP through the mevalonate pathway (*Kampranis and Makris, 2012*). Specifically for forskolin pathway, it seems clear that there is a limitation in flux through one or several of the P450s involved as we encounter accumulation of the intermediates, 300 mg/L and 500 mg/L of compounds **1** and **3a** respectively, compared to forskolin (40 mg/L). Accumulation of only minute amounts of deacetylforskolin shows that *Cf*ACT1-8 is not a limiting step in the pathway. Accumulation of **1** and **3a** intermediates was not observed *in planta* (*Figure 10—figure supplement 1*). This likely signifies that the expression levels of the heterologous CYPs expressed in yeast are not properly balanced or their efficiency and activity can be affected negatively after incooporation into the yeast membrane, while in the native plant host, the pathway exhibits optimized carbon flux and enzymatic efficiency. P450s are notorious difficult to express in high amounts in yeast and recognized as exhibiting rather low $K_{cat}$ values (*Jung et al., 2011*; *Renault et al., 2014*). Hence, to increase forskolin production in the yeast system, efforts should obviously be focused on optimizing *Cf*CYP76AHs expression and enzyme kinetics, specificity

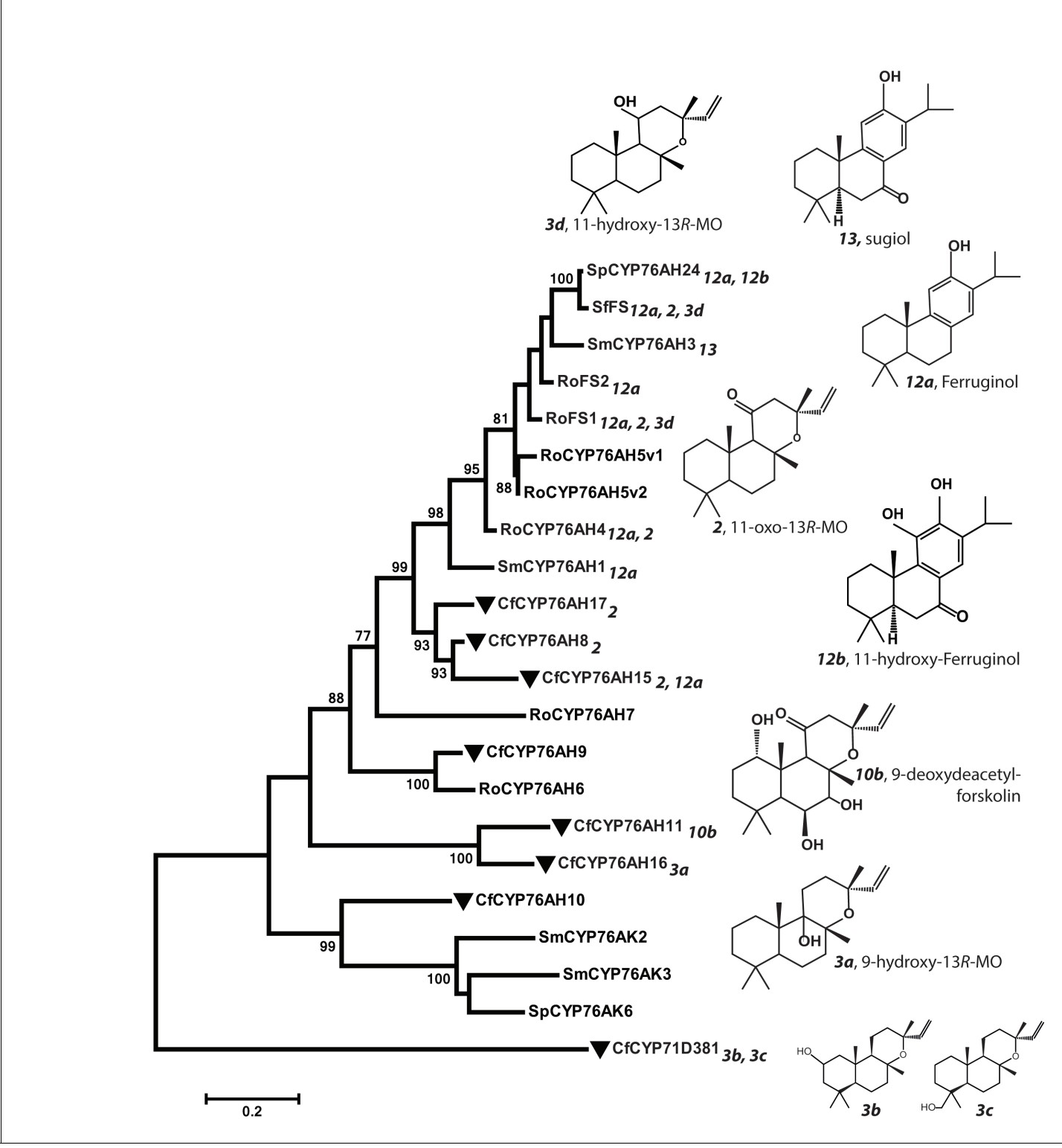

**Figure 11.** Phylogeny of known full-length CYP76AHs. The enzymes used are listed below with their accession numbers or source of publication: *Cf*CYP76AH15, KT382358; *Cf*CYP76AH17, KT382360; *Cf*CYP76AH8, KT382348; *Cf*CYP76AH11, KT382349; *Cf*CYP76AH16, KT382359; *Cf*CYP76AH9, KT382347; *Cf*CYP76AH10, KT382346; *Cf*CYP71D381, KT382342; *Ro*FS1, AJQ30187 (***Božić et al., 2015***); *Sm*CYP76AH3, KR140168 (***Guo et al., 2016***); *Ro*FS2, AJQ30188 (***Božić et al., 2015***); *Sf*FS, AJQ30186 (***Božić et al., 2015***); *Ro*CYP76AH4, (***Zi and Peters, 2013***); *Ro*CYP76AH5v1, (***Zi and Peters, 2013***); *Ro*CYP76AH5v2, (***Zi and Peters, 2013***); *Ro*CYP76AH6, (***Zi and Peters, 2013***); *Ro*CYP76AH7, (***Zi and Peters, 2013***); *Sm*CYP76AH1, AGN04215 (***Guo et al., 2013***); *Sp*CYP76AH24, ALM25796 (***Ignea et al., 2016a***). *Coleus forskohlii* enzymes are indicated by a solid black triangle. *Cf*CYP71D381

*Figure 11 continued on next page*

*Figure 11 continued*

was chosen as a root because it can accept 13*R*-manoyl oxide as a substrate, but does not catalyze the synthesis of forskolin-related products. The number subscripts indicated at each enzyme refer to their respective enzymatic products, the structures of which are given on the right. Only the main products of each enzymes are mentioned. MO stands for manoyl oxide.

and catalytic efficiency as well as pathway scaffolding to facilitate formation of a metabolon which will result in improved pathway flux and efficiency and reduced accumulation of pathway intermediates (*Laursen et al., 2016*). The high forskolin titers already obtained though in the engineered yeast strain highlights the potential to develop a microbial manufacturing platform for efficient and stereo-specific production of forskolin and other labdane terpenoids by fine tuning the biosynthetic pathways.

A yeast-based production platform constitutes a sustainable alternative to traditional crop-based production but the gains always need to be compared to yield improvements obtained by classical or molecular breeding of the traditional host plant (*Graham et al., 2010*). The model-example from the literature and industry toward production of a pharmaceutically relevant terpenoid in *S. cerevisiae* is the sesquiterpenoid artemisinic acid, a pathway intermediate to the antimalarial compound artemisinin (*Paddon et al., 2013*). However, this prominent model example is dependent on a costly organic chemical synthesis component to chemically convert artemisinic acid to artemisinin (*Peplow, 2016*). Recent approaches of engineered de novo production of structurally complex diterpenoids, triterpenoids or alkaloids in microbial systems have also been limited to proof-of-concept studies, expression of partial pathways and sub-milligram yields, highlighting the challenges in synthetic biology to offer an economically realistic and sustainable alternative to isolation of the desired medicinal compounds from medicinal plants bred to produce elevated levels. The constraints to achieve high yields are connected to expression of multiple CYPs and reconstruction of pathways with multiple functionally divergent steps (*Brown et al., 2015*; *Li and Smolke, 2016*; *Zhou et al., 2015*). Strategies addressing these issues are for example the development of synthetic microbial consortia of *S. cerevisiae* and *E. coli*, optimization of CYPS for functional expression in *E. coli*, optimization of interactions between the CYPs and their reductase partner and N-terminal modifications (*Biggs et al., 2016*; *Laursen et al., 2016*; *Vazquez-Albacete et al., 2017*; *Zhou et al., 2015*).

Our current study demonstrates that mining for additional members of the CYP76AH family has the potential to facilitate the assembly of further optimized panels of mixed-species P450s for the biosynthesis of bioactive diterpenoids. This shows the great promise that combinatorial assembly including CYPs outside the CYP76AH subfamily may offer and the opportunity to design production systems for diterpenoids that are currently inaccessible due to their exclusive presence in rare or red-listed plants and to further expand the chemical diversity of diterpenoids to production of compounds currently not known in nature.

# Materials and methods

## Materials

All chemicals including forskolin were acquired from Sigma-Aldrich. An authentic standard of 13*R*-manoyl oxide was prepared as previously described (*Nielsen et al., 2014*). CYP76AH4 (*Zi and Peters, 2013*) was cloned from rosemary plants acquired at a local market in Copenhagen, Denmark. *Ro*FS1 and *Sf*FS (*Božić et al., 2015*) were kindly provided by Dr. Angelos Kanellis (University of Thessaloniki, Greece).

## Chemical synthesis of deacetylforskolin

The deacetylation of forskolin has been achieved previously with the use of methanolic potassium carbonate which can provide 7-desacetylforskolin in 65% yield (*Kosley and Cherill, 1989*). Here, we carried out the deacetylation of forskolin using a solution of methanolic ammonia solution (2M) to afford 7-desacetylforskolin quantitatively. The 1 hr NMR data of the deacetylated forskolin were in agreement with the reported data in literature.

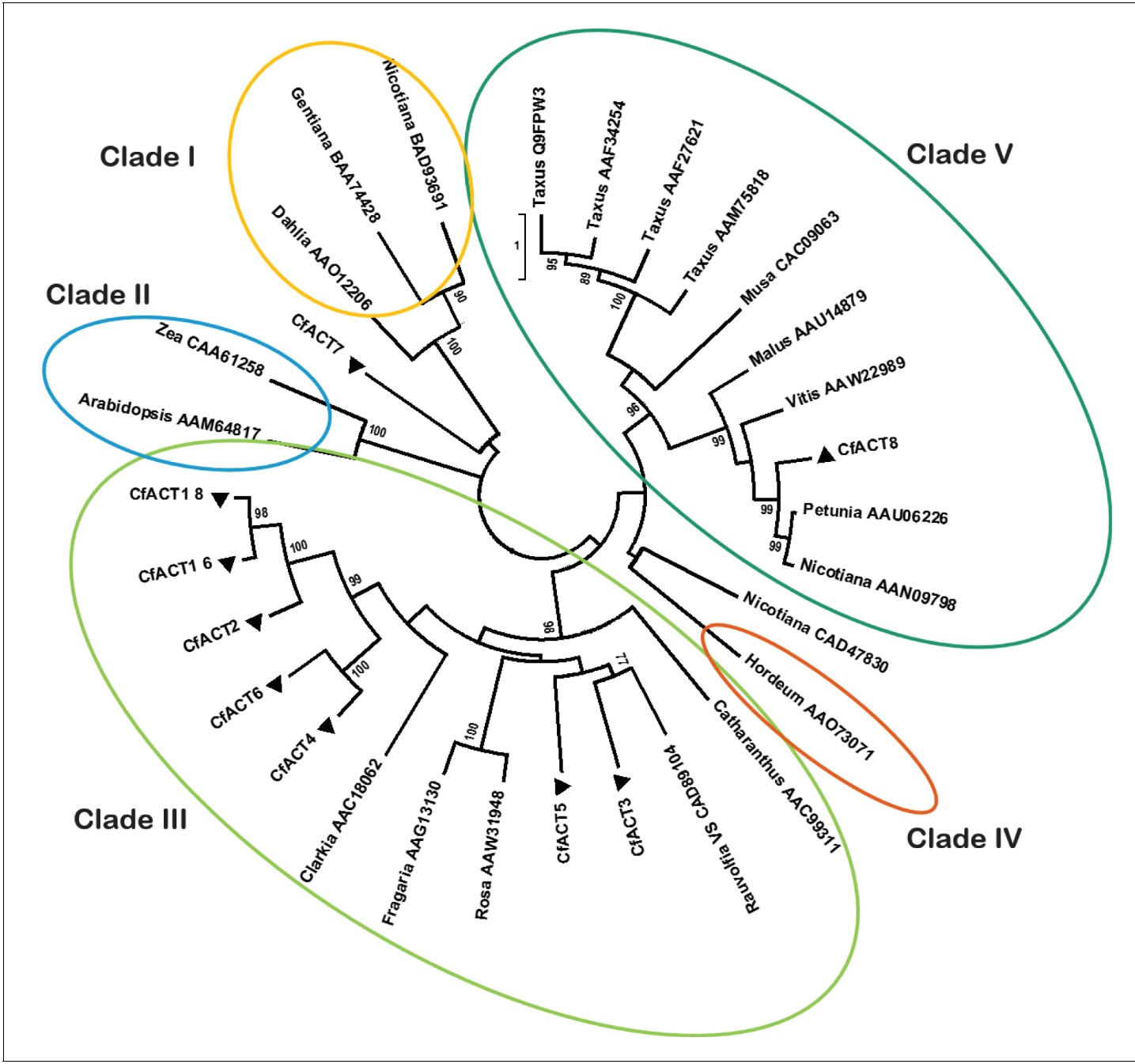

**Figure 12.** Phylogenetic tree of *Cf*ACT encoding candidate genes together with BAHD family acyltransferase representatives from all clades according to *D'Auria (2006)*. Accession numbers of the non-*Coleus forskohlii* selected protein sequences are shown next to the tree taxon names, while *C. forskohlii* peptide accession numbers are provided in *Figure 1—source data 1*. The analysis only includes functionally characterized members. *Coleus forskohlii* enzymes are indicated by a solid black triangle. The majority of the selected *Cf*ACTs belong to Clade III, which includes mainly members which accept a diverse range of hydroxylated substrates and use acetyl-CoA as the main acyl donor (*D'Auria, 2006*). Interestingly, the ACTs known to be involved in Taxol biosynthesis belong to Clade V.

## Transcriptome sequencing

*Coleus forskohlii* root cork total RNA was extracted as previously described (*Pateraki et al., 2014*). RNA was prepared for sequencing using the Illumina TruSeq sample preparation kit v2 using poly-A selection (Illumina San Diego, USA). The fragments were clustered on cBot and sequenced with

paired ends (2 × 100 bp) on a HiSeq 2500 (Illumina San Diego, USA) according to the manufacturer's instructions. A total 106.2 million read-pairs were generated. Adaptor sequences were removed from raw reads and reads were trimmed at the ends to phred score 20, using the fastq-mcf tool from ea-utils (https://code.google.com/p/ea-utils/). Processed reads were assembled using Trinity (r2013-02-16) resulting in a total of 263,652 assembled putative transcripts. Transcript abundance estimation was performed using RSEM and the scripts provided with Trinity. Likewise, the putative coding sequences were predicted using the TransDecoder scripts from Trinity. The high-throughput RNA sequences reported here have been submitted to the short read archive (SRA) at the NCBI [accession no SAMN06013363].

## Identification and cloning of genes involved in forskolin biosynthesis

Mining of the *C. forskohlii* transcriptome database was performed as previously described (*Zerbe et al., 2013*) using tBLASTx software and known CYP or acetyl transferase (ACT) sequences as query. The identified contigs were amplified from single stranded cDNA generated from root cork total RNA using the 'SuperScript III First-Strand Synthesis System for RT-PCR' (Invitrogen) and oligo-dT primer. Cloning of the putative CYP and ACT cDNAs was achieved after PCR amplification using gene-specific primers (*Figure 1—source data 3*) that were designed based on the in silico sequences of the identified CYP and ACT contigs (*Figure 1—source data 1*). PCR products were cloned into the pJET1.2 vector and verified by sequencing. For the identified non-full-length cDNAs, (*Cf*CYP76AH15, *Cf*CYP76AH16 and *Cf*ACT1-8), full-length transcripts were obtained using 5'RACE techniques.

## Phylogenetic analysis of CYP76AH and ACT candidate enzymes

Amino acid sequences of the CYP76AHs of *C. forskohlii* and of currently available CYP76AHs from other plant species were used to construct a phylogenetic tree to infer the evolutionary history of these enzymes. Sequences were retrieved from the current work, GenBank database (http://www.ncbi.nlm.nih.gov/) and original articles. Peptide alignments were performed using the MUSCLE program included in the MEGA6 software. Phylogenetic analyses were performed by the Maximum Likelihood method based on the Dayhoff matrix-based model with 'uniform rates' (*Schwarz and Dayhoff, 1979*) and using all sites with 'Nearest-Neighbor-Interchange' heuristic method conducted using MEGA6 software (*Tamura et al., 2013*). The tree is drawn to scale, with branch lengths measured in the number of substitutions per site. Bootstrap values shown in % were inferred from 1000 replicates. Branches supported by bootstrap values higher than 75% are shown.

For the phylogenetic analyses of *Cf*ACT candidates, the identified sequences from *C. forskohlii* were analyzed together with BAHD family acyltransferase representatives from all clades (*D'Auria, 2006*). The analysis includes only functionally characterized members. The analysis was inferred by using the Maximum Likelihood method based on the Dayhoff matrix-based model with 'uniform rates' and using all sites (*Schwarz and Dayhoff, 1979*). Initial trees for the heuristic search were obtained automatically by applying Neighbor-Join and BioNJ algorithms to a matrix of pairwise distances estimated using a JTT model, and then selecting the topology with superior log likelihood value. The tree is drawn to scale, with branch lengths measured in the number of substitutions per site. Evolutionary analyses were conducted using MEGA6 software (*Tamura et al., 2013*).

## RNA extraction and quantitative real-time PCR

Total RNAs from *C. forskohlii* tissues were extracted as previously described (*Pateraki et al., 2014*) and digested by DNase I on-column. The integrity of the RNA samples was evaluated using the RNA-nano assay using an Agilent 2100 Bioanalyser (Agilent Technologies). First-strand cDNAs were synthesized from 0.5 μg of total RNA from an oligo-dT primer, using the 'SuperScript III First-Strand Synthesis System for RT-PCR' (Invitrogen). The resulting cDNAs were diluted 10-fold. For the qRT-PCR reactions, gene specific primers were used (*Figure 1—source data 3*) with Maxima SYBR Green/Fluorescein qPCR Master Mix (Fermentas) on a Rotor-Gene Q cycler (Qiagen) using the following cycling parameters: 95°C for 7 min, 35 cycles of 95°C for 15 s, 60°C for 30 s and 72°C for 30 s followed by a melting curve cycle from 60°C to 90°C. Eukaryotic initiation factor 4A (TIF4a) and Elongation Factor 1A (EF1a) were used as reference genes because they showed the lowest variation across different tissues (*Pateraki et al., 2014*). No statistically significant differences were observed

between the results obtained using the two different reference genes and the results presented were normalized based on TIF4a. Relative transcript abundance was calculated as the mean of three biological replications obtained using three different *C. forskohlii* plants, while the reactions were performed in three technical replicates. Amplification efficiency was calculated with the 'Real Time PCR Miner' (http://www.miner.ewindup.info/Version2). Efficiency-corrected $\Delta$CT values were used to quantify relative differences in target gene transcript accumulation. Primer specificity was assessed by agarose gel analysis and sequencing of amplicons from representative reactions, as well as from melting curve analysis of every reaction.

## Functional characterization of *Coleus forskohlii* cytochrome P450 enzymes (*Cf*CYPs) by transient expression in *N. benthamiana*

Functional characterization of the selected candidate genes was obtained using transient expression in *N. benthamiana* which offers optimal native plant protein translation and processing, convenient rapid and optional combinatorial co-expression of multiple genes from independent vectors, native subcellular location of diTPS and CYPs as well as the availability of an endogenous native pathway providing GGPP.

For the functional characterization of *C. forskohlii* selected CYPs and testing of their ability to hydroxylate 13*R*-manoyl oxide (**1**), genes encoding candidate *Cf*CYPs were transiently co-expressed in *N. benthamiana* leaves together with *C. forskohlii* enzymes boosting the formation of **1**, namely 1-deoxy-D-xylulose 5-phosphate synthase (*Cf*DXS), geranylgeranyl diphosphate synthase (*Cf*GGPPS), *Cf*TPS2 and *Cf*TPS3 (*Pateraki et al., 2014*). Transient expression in *N. benthamiana* was performed as previously described (*Bach et al., 2014*). *Cf*CYP cDNAs selected for functional testing were subcloned into pCAMBIA130035Su by USER cloning (*Nour-Eldin et al., 2010*). Vectors carrying the selected cDNAs were then transformed into the agrobacterium strain AGL-1-GC3850 (*Bach et al., 2014*). For the agro-infiltration, the OD of the agrobacterium cultures of all transformed agrobacteria strains was normalized to $OD_{600}$ = 1. Different combinations prepared from equal volumes of each culture of transformed agrobacterium strains expressing individual genes encoding *Cf*CYPs or 13*R*-manoyl oxide biosynthetic enzymes were infiltrated into the leaves of 4–6 weeks old *N. benthamiana* plants. Controls encompassing expression of the *Cf*CYP encoding genes without the 13*R*-manoyl oxide biosynthetic genes were used to assess the possibility of *Cf*CYPs cross-reactivity with tobacco endogenous diterpenoids.

Metabolites from transgenic *N. benthamiana* leaves were extracted by hexane and 85% MetOH and were analyzed by GC-MS and LC-MS-qTOF, respectively. LC-MS-qTOF analysis was performed on the system that was comprised of an Agilent G1312B SL binary pump, Agilent G1367C SL WP autosampler, Agilent G1316B column oven, Agilent G1315C Starlight DAD detector and Bruker microTOF II Mass Spectrometer using Electron Spray Ionization (ESI). Samples were separated on a Synergi 2.5 mm Fusion-RP C18 column (50 × 3.2 mm i.d., Phenomenex Inc., Torrance, CA, USA) at a flow rate of 0.2 mL/min with column temperature held at 25°C. The mobile phase consisted of water with 0.1% formic acid (v/v; solvent A) and 80% acetonitrile with 0.1% formic acid (v/v; solvent B). The gradient program was 0 min, 60% B; 25 min, 98% B; 31 min, 98% B; 32 min, 60% B; 42 min, 60% B. Mass spectra were acquired in positive ion mode using a drying temperature of 200°C, a nebulizer pressure of 3.0 bar, and a drying gas flow of 7 L/min (*Luo et al., 2016*).

GC-MS analysis was performed on a Shimadzu GCMS-QP2010 Ultra using a 3 Agilent HP-5MS column (20 m × 0.180 mm i.d., 0.18 μm film thickness). Injection volume and temperature was set to 1 μL and 250°C. GC program: 60°C for 1 min, ramp at rate 30°C min-1 to 180°C, ramp at rate 10°C min-1 to 290°C, ramp at rate 30°C min-1 to 320°C and hold for 2 min. $H_2$ was used as carrier gas. Transfer line temperature was set to 280°C and electron impact (EI) was used as ionization method in the mass spectrometer (MS) with the ion source temperature and voltage set to 300°C and 70 eV. MS spectra were recorded from 50 m/z to 400 m/z.

## Large-scale biosynthesis of oxygenated 13*R*-manoyl oxide compounds for NMR analysis

11-Oxo-13*R*-manoyl oxide (**2**), 2-hydroxy-13*R*-manoyl oxide (**3b**), 19-hydroxy-13*R*-manoyl oxide (**3c**), 1,11-dihydroxy-13*R*-manoyl oxide (**5d**), 1,9-deoxydeacetylforskolin (**7** hr) and 9-deoxydeacetylforskolin (**10b**) were produced using the biosynthetic scheme described above by large-scale expression of

the relevant gene combinations in *N. benthamiana* (*Andersen-Ranberg et al., 2016*). *CYP76AH8* was expressed to obtain biosynthesis of compound (**2**), *CYP71D381* to obtain (**3b**) and (**3c**), whereas combined expression of *CfCYP76AH8* and *CfCYP76AH11* afforded (**5d**), (**7 hr**) and (**10b**). For the large-scale experiments, agroinfiltration was performed by vacuum infiltration. For biosynthesis of each compound in amounts sufficient for NMR analysis, 100–200 g of fresh weight *N. benthamiana* leaf material were harvested 7 days after infiltration. Leaf material chopped into small pieces was extracted using 0.5 L of *n*-hexane. The solvent was thrice evaporated and recovered by rotor evapo-ration for repeated extraction of the same leaf material. Concentrated extracts from *N. benthamiana* leaves biosynthesizing target compounds were subjected to solid phase extraction (SPE), using silica gel with *n*-hexane and ethyl acetate mixtures (100:0, 99:1, 98:2, 94:6, 92:8, 88:12 (v/v)) in steps of 100 mL. Fractions containing diterpenoids were identified by GC-MS. Diterpenoid-containing frac-tions were combined and the solvent removed by rotor evaporation and the samples resuspended in 1 mL *n*-hexane affording a crude fraction for further purification.

## Isolation of oxygenated 13*R*-manoyl oxide-derived diterpenoids from *N. benthamiana* extracts

Final isolation of individual diterpenoids was achieved using an Agilent 7890B GC installed with an Agilent 5977A MSD, GERSTEL Preparative Fraction Collector (PFC) AT 6890/7890 and a GERSTEL CIS 4C Bundle injection port. Separation was carried out using a RESTEK Rtx-5 column (30 m × 0.53 mm i.d.×1 μm $d_f$) with $H_2$ as carrier gas. At the column outlet, a splitter was mounted with a split vent ratio of 1:100 to the MS and the PFC, respectively. Sufficient amounts of oxidized 13*R*-manoyl oxide-derived diterpenoids for NMR analysis (0.5–1 mg) were obtained by 100 repeated injections of 5 μL extract aliquots. Injection port was set in solvent vent mode with a $H_2$ gas flow of 100 mL/min until 0.17 min, combined with a sample injection speed of 1.5 mL/min. Purge flow was set to 3 mL/min from 0.17 min to 2.17 min. Injection temperature was held at 40°C for 0.1 min followed by ramping at 12°C/s until 320°C, which was held for 2 min. Column flow was set to 7.5 mL, which was held constant throughout the GC program. The GC program was set to hold at 60°C for 1 min, ramp 30°C/min to 220°C, ramp 2°C/min to 250°C and a final ramp of 30°C/min to 220°C, which was held for 2 min. Temperature of the transfer line from GC to PFC and the PFC itself was set to 250°C. The PFC was customized to collect the peaks of **2**, **3b**, **3c**, **5d** and **7** hr, and **10b** by their retention time identified by the MS. The MS for monitoring the PFC purification was set in scan mode from *m/z* 35 to *m/z* 500, with a threshold of 150 ion counts. Solvent cut-off was set to 4 min and the temper-ature of the MS source and the MS quadropole - to 300°C and 150°C, respectively.

## Large-scale biosynthesis in *S. cereviseae* and isolation of 9-hydroxy-13*R*-manoyl oxide (3a) for NMR analysis

For biosynthesis of **3a** in amounts sufficient for NMR analysis, the *S. cerevisiae* strain EFSC4494 car-rying chromosomally integrated *CYP76AH16* was inoculated into a pre-culture of 5 mL selective media (SC-Ura) and grown for 16 hr at 30°C and 400 rpm. A 1 mL aliquot of the pre-culture was used for inoculation of 100 mL non-selective media (SC) and grown in a 500 mL Erlenmeyer shake-flask for 120 hr at 30°C with horizontal shaking at 180 rpm. Following addition of 100 mL UV-grade 99.9% ethanol and maintenance of the sample at 60°C for 20 min, 200 mL of *n*-hexane was added and the sample shaken for 2 hr at room temperature. The hexane phase was concentrated by rotor evaporation and subjected to column chromatography (dual layer Florisil/$Na_2SO_4$6 mL PP SPE TUBE, Supelco Analytical) with a gradient composed of *n*-hexane and 1–15% ethyl acetate.

## Metabolic engineering of *S. cerevisiae* for the production of forskolin

All genes selected for functional expression in *S. cerevisiae* were codon optimized for efficient expression in *S. cerevisiae*, and purchased as DNA STRINGS (Geneart, LifeTechnologies). Genomic integration was chosen over expression via episomal plasmids to favor simultaneous expression of a number of genes as well as to enable the use of selection marker recycling (*Jensen et al., 2014*). The 13*R*-manoyl oxide producing *S. cerevisiae* strain EFSC4498 (*Andersen-Ranberg et al., 2016*) was used to test all the selected gene combinations for their ability of affording synthesis of forskolin and intermediate products. All genes were cloned into yeast genome integration plasmids by the USER technique (*Nour-Eldin et al., 2010*) targeting incorporation into site XI-2 (*Mikkelsen et al.,*

*2012*). Transformants were verified by PCR on genomic DNA for correct insertion of heterologous genes and grown and tested in 96-deep-well plates (*Andersen-Ranberg et al., 2016*). The yeast strain found to produce the highest amount of forskolin and which exhibited stability through several cultivation rounds (EVST21543) was selected for cultivation for 140 hr in a 5 L fermenter using minimal medium and glucose-limited conditions. Forskolin production was monitored using withdrawn culture aliquots. Forskolin was extracted from the mixture of yeast cells and culture broth using 85% methanol and incubation for 20 min at 75°C and the extract centrifuged (10,000 g for 5 min) to precipitate yeast debris. The supernatant obtained was used after filtration for LC-MS analysis and forskolin quantification.

### Forskolin quantification synthesized from yeast strain EVST21543 growing in the fermenter and comparison to *C. forskohlii* root extract

For forskolin quantification, aliquots of the yeast samples (with broth) collected at specific time points (*Figure 3*) were combined with methanol to give a concentration of 85% methanol, incubated at 75°C for 20 min, filtered and then analyzed by LC-MS. Quantification was based on a standard calibration curve of forskolin purchased from Sigma-Aldrich. An Ultimate 3000 UHPLC$^+$ Focused system (Dionex Corporation, Sunnyvale, CA) coupled to a Bruker Compact ESI-QTOF-MS (Bruker Daltonik, Bremen, Germany) was used to quantify forskolin. Samples were separated on a Kinetex XB-C18 column (100 × 2.1 mm i.d., 1.7 µm particle size, 100 Å pore size; Phenomenex Inc., Torrance, CA) maintained at 40°C with a flow rate of 0.3 mL/min and mobile phase consisting of 0.05% (v/v) formic acid in water (solvent A) and 0.05% (v/v) formic acid in acetonitrile (solvent B). The gradient LC method used for quantification was as follows: solvent B was held at 20% for 30 s, then ramped to 100% over 8.5 min, held at 100% for 2 min, decreased to 20% over 30 s and held for 3.5 min to give an overall run time of 15 min. The ESI source parameters were as follows: capillary voltage, 4500 V; nebulizer pressure 1.2 bar; dry gas flow, 8 l/min; dry gas temperature, 250°C. The QTOF-MS was operated in MS only mode with collision cell energy of 7 eV and collision cell RF of 500 Vpp. Ions were monitored in the positive mode over a range of 50–1300 *m/z* and spectra collected at a rate of 2 Hz. For comparison of yeast profiles to *C. forskohlii* root extract, roots were grinded and then extracted with 85% methanol, incubated at 75°C for 30 min, filtered and analyzed by LC-MS. Analysis was performed as described for forskolin quantification but with the following gradient method: 20% B for 1 min, increased to 100% B over 22 min and then returned to 20% B in 0.5 min and held for 4 min.

### Quantification of 13*R*-manoyl oxide (1) and 9-hydroxy-13*R*-manoyl oxide (3a) synthesized from yeast strain EVST21543 growing in the fermenter

Yeast samples were collected at specific time points (*Figure 10*) and samples kept at −20°C in glass vials. For diterpenoid extraction, 500 µL of *n*-hexane was added to 500 µL yeast broth, shaken for 1 hr at room temperature and separated into two phases by centrifugation at 2500 rpm and stored overnight at 4°C. The hexane phase was then diluted 10 times and run on a SCION 436 GC-FID (Bruker). Sample (1 µL) was injected in splitless mode at 280°C. The GC-program was as follows: 60°C for 1 min, ramp at 20°C/min to 160°C, ramp at 10°C/min from 160°C to 240°C, ramp at 20°C/min from 240°C to 320°C, hold at 320°C for 2 min. H$_2$ was used as carrier gas with a linear flow of 50 mL/min. The FID was set at 300°C, with N$_2$ flow of 25 mL/min, H$_2$ at 30 mL/min and air at 300 mL/min. Data sampling rate was 10 Hz. Compounds **1** and **3a** were identified by comparing the retention time with an authentic standard and quantification was based on FID peak area and a standard curve of **1**.

### NMR analysis

All NMR experiments were recorded at 300 K in CDCl$_3$ using a Bruker Avance III 600 MHz NMR spectrometer ([1]H operating frequency 600.13 MHz) equipped with a Bruker SampleJet sample changer and a cryogenically cooled gradient inverse triple-resonance 1.7 mm TCI probe-head (Bruker Biospin, Rheinstetten, Germany). The experiments were acquired in automation (temperature equilibration to 300 K, optimization of lock parameters, gradient shimming, and setting of receiver gain). Both one-dimensional [1]H and [13]C spectra were acquired with 30°-pulses and 64k data points. The [1]H spectra were recorded with 3.66 s inter-pulse intervals and the FID was multiplied

with an exponential function corresponding to line-broadening of 0.3 Hz prior to Fourier transform. An acquisition time of 0.9 s were used for the $^{13}$C experiments with an additional relaxation delay of 2.0 s. Protons were decoupled during acquisition using waltz16 composite pulse sequence. Backward linear prediction was used to correct the first complex data points of $^{13}$C FIDs before zero-filling to 128k data points and application of exponential window function with a line-broadening factor of 1.0 Hz. Two-dimensional homo- and heteronuclear experiments were acquired with 2048 data points in the direct dimension and 128 (DQF-COSY and HMBC) or 256 (multiplicity edited HSQC and phase sensitive NOESY) data points in the indirect dimension; with spectral widths optimized from the corresponding $^1$H spectra. The HMBC and HSQC experiments were optimized for $^nJ_{H,C}$ = 10 Hz and $^1J_{H,C}$ = 145 Hz, respectively. Acquisition and processing of NMR data were performed using Topspin ver. 3.0 (Bruker Biospin GmbH), and IconNMR ver. 4.2 (Bruker Biospin GmbH) was used for controlling automated sample change and acquisition.

## Acknowledgements

We thank Dr. Sotirios Kampranis for providing ferruginol and 11$\beta$-hydroxy-13$R$-manoyl oxide reference compounds and Dr. Angelos Kanellis for providing RoFS1 and SfFS cDNAs. This work was supported by the VILLUM research center"Plant Plasticity' (BLM), the Center for Synthetic Biology 'bioSYNergy' supported by the UCPH Excellence Program for Interdisciplinary Research (BLM), the Novo Nordisk Foundation (BjH), an ERC Advanced Grant to BLM (ERC-2012-ADG_20120314), individual Marie Skłodowska-Curie fellowships to IP and AMH. BjH is in part supported by the DOE Great Lakes Bioenergy Research Center (DOE Office of Science BER DE-FC02-07ER64494) and gratefully acknowledges the Strategic Partnership Grant (15-SPG-Full-3101), MSU Foundation, startup funding from the Department of Molecular Biology and Biochemistry, Michigan State University and support from Michigan State University AgBioResearch (MICL02454).

## Additional information

### Competing interests

IP, JA-R, BLM: Filed international patent 600 applications (PCT/DK2015/050020) covering 'Biosynthesis of forskolin and related compounds. NBJ: Filed international patent 600 applications (PCT/DK2015/050020) covering 'Biosynthesis of forskolin and related compounds Employee of Evolva SA. JH: Employee of Evolva SA. BH: Filed international patent 600 applications (PCT/DK2015/050020) covering 'Biosynthesis of forskolin and related compounds'. The other authors declare that no competing interests exist.

### Funding

| Funder | Grant reference number | Author |
|---|---|---|
| European Commission | ERC-2012-ADG_20120314 | Irini Pateraki |
| Villum Fonden | | Birger Lindberg Møller |
| Novo Nordisk | | Birger Lindberg Møller |
| Biological and Environmental Research - DOE Office of Science | DE-FC02-07ER64494 | Björn Hamberger |
| Michigan State University | Strategic Partnership Grant (15-SPG-Full-3101) | Björn Hamberger |
| Michigan State University | MICL02454 | Björn Hamberger |

The funders had no role in study design, data collection and interpretation, or the decision to submit the work for publication.

### Author contributions

IP, Conceptualization, Data curation, Investigation, Methodology, Writing—original draft, Project administration; JA-R, AMH, Data curation, Formal analysis, Investigation, Methodology, Writing—review and editing; NBJ, Data curation, Formal analysis, Investigation, Methodology; SGW, Data

curation, Formal analysis, Methodology; VF, Data curation, Formal analysis, Investigation; BHal, Formal analysis, Methodology; BHam, Formal analysis; MSM, Methodology, Writing—review and editing; CEO, Data curation; DS, Data curation, Methodology, Writing—review and editing; JH, Supervision, Methodology; BLM, Funding acquisition, Writing—original draft, Writing—review and editing; BH, Conceptualization, Writing—review and editing

## Author ORCIDs
Irini Pateraki, http://orcid.org/0000-0002-7526-2334
Allison Maree Heskes, http://orcid.org/0000-0002-2732-5185
Björn Hamberger, http://orcid.org/0000-0003-1249-1807

## Additional files

### Supplementary files
• Supplementary file 1. Overview of 13*R*-manoyl oxide-derived diterpenoids identified in *N. benthamiana*, expressing combinations of *C. forskohlii* genes encoding CYPs and acetyltransferases together with genes encoding the required enzymes for biosynthesis of 13*R*-manoyl oxide (*Cf*DXS, *Cf*GGPPs, *Cf*TP2 and *Cf*TPS3). GC-MS and LC-qTOF-MS chromatograms of the identified diterpenoids are shown in previous figures.

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
