## [Decision Letter]

Thank you for submitting your article "Total biosynthesis of the cyclic AMP booster forskolin from *Coleus forskohlii*" for consideration by *eLife*. Your article has been reviewed by two peer reviewers, and the evaluation has been overseen by Joerg Bohlmann as the Reviewing Editor and Ian Baldwin as the Senior Editor. The reviewers have opted to remain anonymous.

The reviewers have discussed the reviews with one another and the Reviewing Editor, and they agreed on all of the requests for revisions. Both reviewers were quite positive about the work presented in your paper. Their specific comments and requests for revisions are below.

Reviewer #1:

The manuscript by Pateraki et al. reports on the identification and characterization of genes (and encoded enzymes) for the biosynthesis of forskolin, a diterpenoid natural product accumulated in the roots of *Coleus forskohlii*. The research team took advantage of *Nicotiana benthamiana* as a host for the expression of combinations of candidate genes that were identified based on analyses of appropriate transcriptome data sets acquired for various *Coleus* tissues. The research approach is very innovative and the results are exciting. However, there are a few issues that need to be addressed:

While the manuscript is generally well written, there are several inaccuracies that need to be addressed:

Introduction, first paragraph: Pateraki et al. (2015) may indeed mention that 50,000 terpenoids of plant origin have been characterized, but I wonder who counted those and how? I do not think repeating made-up numbers is useful.

Introduction, first paragraph: Who says that terpenoids are the oldest class of plant specialized metabolites and why?

“Hence, commercially available forskolin is extracted from *C. forskohlii* roots and purified from a mixture of over structurally related abietane and epoxylabdane diterpenoids to give an average yield of 0.03% of root dry weight (Alasbahi et al., 2010a; Asada et al., 2012)”: The way this sentence is written, it would appear that the yield of forskolin after purification is 0.03% of root dry weight. I don't think that is correct. Please rephrase.

Subsection “Discovery of multifunctional cytochromes P450 in *Coleus forskohlii* producing a multitude of 13R-manoyl oxide-derived diterpenoids and identification of a biosynthetic pathway for forskolin”, sixth paragraph and elsewhere: in mass spectrometry, the most commonly used unit is the mass over charge ratio (the technology does not report accurate mass directly).

Figure 2 and corresponding narrative: some of the metabolites were identified, at least in part, by GC-MS using a quadrupole instrument. The mass accuracy of this equipment is not sufficient to deduce a molecular formula (as claimed in the legend). Please rephrase.

General remarks:

Entire Results section: The Results section contains many statements that directly discuss the results. It would be preferable to move these sentences to the Discussion section.

Figure 2: Many of the oxygenated products detected in extracts of transgenic *N. benthamiana* remain unidentified. The authors acknowledge challenges but it would be good to discuss the issue in more depth (why is it difficult to obtain larger quantities? What can be done?…).

The maximum yield of forskolin in an engineered yeast strain was 40 mg/L. What would be the target for this to be commercially viable? What are the critical challenges and which approaches can be suggested to overcome such barriers?

Additional data:

Subsection “Discovery of multifunctional cytochromes P450 in *Coleus forskohlii* producing a multitude of 13R-manoyl oxide-derived diterpenoids and identification of a biosynthetic pathway for forskolin”, last paragraph: the authors demonstrate that the enzymes encoded by candidate genes from *C. forskohlii* produce a variety of labdane products. A more quantitative assessment would be highly desirable. The authors have authentic standards for many terpenoid pathway intermediates and the development of a quantitative method should thus not be overly challenging. Which intermediates accumulate in vivo? Can this be discussed in the context of the biochemistry as elucidated here? What are the implications for pathway control?

Reviewer #2:

In this manuscript, Pateraki et al. report the systematic discovery and characterization of numerous P450s and acyltransferases that catalyze transformations to functionalize the forskolin precursor 3-manoyloxide into diverse terpenoids. Three of the P450s and one acyltransferase are found to be the minimal set needed to convert manoyloxide into forskolin, completing the pathway to this important natural product.

The relevance of this work is nicely highlighted by the authors' demonstration that the pathway can be engineered into yeast, yielding the production of forskolin. This is a well done study that represents an exciting advance for the engineering of plant biosynthetic pathways. It also has implications for how plants produce suites of related molecules, given that many of the enzymes studied have overlapping functions. This paper will be of broad interest.

Requested Revisions:

For each figure where an extracted ion chromatogram is shown (as is done in Figure 9), somewhere in the figure or caption the mass ions extracted to generate the EIC should be listed. As a side note, in Figure 9 it would be helpful to use consistent formatting and list the enzymes expressed to generate each of the chromatograms (in the figure itself, rather than the caption).

One section of the results describes monooxygenase activity of the CYP67AH subfamily towards other terpene scaffolds. While the study of substrate promiscuity of the 76AH subfamily of enzymes is relevant, this part is dense and difficult to follow. A sentence or two of introductory text would help orient the reader here.

The authors might consider adding to the phylogenetic tree shown in Figure 8 the demonstrated biochemical activities of the relevant enzymes, to help the reader understand the relationship between the phylogenetic analysis and the biochemistry.

In the NMR data, it would useful for the community if the actual NMR spectra are shown in addition to the tabulated chemical shifts shown in Table 1 and Table 2.

---

## [Author Response]

[…] Reviewer #1:

[…] While the manuscript is generally well written, there are several inaccuracies that need to be addressed:

Introduction, first paragraph: Pateraki et al. (2015) may indeed mention that 50,000 terpenoids of plant origin have been characterized, but I wonder who counted those and how? I do not think repeating made-up numbers is useful.

The number of identified terpenoids we mention has been found in the “Dictionary of natural products” (http://dnp.chemnetbase.com/). We have now included the reference in the first paragraph of the Introduction. More detailed analysis of that can be found in our previous article by: Pateraki I, Heskes AM, Hamberger B (2015). Cytochromes P450 for Terpene Functionalisation and Metabolic Engineering. Biotechnology of Isoprenoids. J Schrader, J Bohlmann. Cham, Springer International Publishing, 10.1007/10_2014_301: 107-139.

Introduction, first paragraph: Who says that terpenoids are the oldest class of plant specialized metabolites and why?

Our idea that terpenoids are the oldest class of specialized metabolites is rooted in the idea that evolution of terpenoids, in the form of sterols, was a critical factor for the evolvement of life in the form we know it today, because terpenoids contributed in formation of primitive membranes according to the article: Ourisson G, Nakatani Y. 1994. The terpenoid theory of the origin of cellular life: the evolution of terpenoids to cholesterol. Chemistry & Biology1:11-

23. We understand though that this argumentation is not about “specialized metabolites” but rather about terpenoids participating in basic metabolism. For that reason, after reviewer’s comment, we have deleted that phrase from the text.

*“Hence, commercially available forskolin is extracted from C. forskohlii roots and purified from a mixture of over structurally related abietane and epoxylabdane diterpenoids to give an average yield of 0.03% of root dry weight (Alasbahi et al., 2010a; Asada et al., 2012)”: The way this sentence is written, it would appear that the yield of forskolin after purification is 0.03% of root dry weight. I don't think that is correct. Please rephrase.*

Thank you for pointing it out, this is a very useful comment. We have rephrased the sentence and added a more recent article showing forskolin content in plant roots (Introduction, second paragraph).

Subsection “Discovery of multifunctional cytochromes P450 in Coleus forskohlii producing a multitude of 13R-manoyl oxide-derived diterpenoids and identification of a biosynthetic pathway for forskolin”, sixth paragraph and elsewhere: in mass spectrometry, the most commonly used unit is the mass over charge ratio (the technology does not report accurate mass directly).

We would like to thank the reviewer for that comment, as it is absolutely correct. We have changed the “accurate mass” with “mass to charge ratio *(m/z)”* (subsection “Discovery of multifunctional cytochromes P450 in *Coleus forskohlii* producing a multitude of 13R-manoyl oxide-derived diterpenoids and identification of a biosynthetic pathway for forskolin”, last paragraph).

Figure 2 and corresponding narrative: some of the metabolites were identified, at least in part, by GC-MS using a quadrupole instrument. The mass accuracy of this equipment is not sufficient to deduce a molecular formula (as claimed in the legend). Please rephrase.

We have changed the legend of Figure 2, according to reviewer’s suggestion. We hope that it is now clear that the identification of the structure of several of the metabolites mentioned in this figure has been carried out by NMR or LC-qTOF-MS.

General remarks:

Entire Results section: The Results section contains many statements that directly discuss the results. It would be preferable to move these sentences to the Discussion section.

To comply with your request we have moved paragraphs from the Results to the Discussion section. The meaning and the context of the text have remained as before.

Figure 2: Many of the oxygenated products detected in extracts of transgenic N. benthamiana remain unidentified. The authors acknowledge challenges but it would be good to discuss the issue in more depth (why is it difficult to obtain larger quantities? What can be done?).

To answer the above questions, a paragraph has been added to the relevant section (subsection “Discovery of multifunctional cytochromes P450 in *Coleus forskohlii* producing a multitude of 13R-manoyl oxide-derived diterpenoids and identification of a biosynthetic pathway for forskolin”, second paragraph). We agree with the reviewer that this is an important point to address, but it is an issue difficult to tackle mainly due to the complexity (many of the peaks are overlapping for example) of the resulting terpenoid profile and the instability of many of these molecules. Additionally, the system we are using in this work for enzyme characterization (transient expression in tobacco leaves) does not provide substantial plant material to perform extensive terpenoids purification in high amounts that would be necessary for NMR studies, for example.

The maximum yield of forskolin in an engineered yeast strain was 40 mg/L. What would be the target for this to be commercially viable? What are the critical challenges and which approaches can be suggested to overcome such barriers?

Additional data:

Subsection “Discovery of multifunctional cytochromes P450 in Coleus forskohlii producing a multitude of 13R-manoyl oxide-derived diterpenoids and identification of a biosynthetic pathway for forskolin”, last paragraph: the authors demonstrate that the enzymes encoded by candidate genes from C. forskohlii produce a variety of labdane products. A more quantitative assessment would be highly desirable. The authors have authentic standards for many terpenoid pathway intermediates and the development of a quantitative method should thus not be overly challenging. Which intermediates accumulate in vivo? Can this be discussed in the context of the biochemistry as elucidated here? What are the implications for pathway control?

To respond to the comment of the reviewer, we have added a figure (Figure 12—figure supplement 1) demonstrating the complexity of the chemical profile of the *C. forskohlii* root in comparison to the yeast forskolin-producing strain. Absolute quantification of the terpenoids of plant’s root does not signify something substantial as forskolin content varies quite dramatically (from 0.013 % to 0.728 % as stated in the manuscript) according to soil and climatic conditions and dependent on the *C. forskohlii* variety used. In general, the most important conclusion that can be deduced from the data set is that in the plant we do not see the pathway intermediates observed in the yeast strain. This would imply that the pathway in its native plant host exhibits a more optimized or channeled carbon flux. In their natural host environment, the CYPs are more efficient/specific towards forskolin production and maybe even towards other end-products that we are not able to identify here. A paragraph discussing these issues has been added at the Results section (subsection “Engineering of the entire pathway of forskolin in Saccharomyces cerevisiae”, last paragraph) and in the Discussion (eighth paragraph).

Reviewer #2:

[…] Requested Revisions:

For each figure where an extracted ion chromatogram is shown (as is done in Figure 9), somewhere in the figure or caption the mass ions extracted to generate the EIC should be listed. As a side note, in Figure 9 it would be helpful to use consistent formatting and list the enzymes expressed to generate each of the chromatograms (in the figure itself, rather than the caption).

The only figure showing EIC is Figure 9. This specific figure has been converted according to the reviewer’s suggestions.

One section of the results describes monooxygenase activity of the CYP67AH subfamily towards other terpene scaffolds. While the study of substrate promiscuity of the 76AH subfamily of enzymes is relevant, this part is dense and difficult to follow. A sentence or two of introductory text would help orient the reader here.

According to reviewer’s suggestion, we have added a few sentences in the text regarding activity of CYP76AHs, subsection “Monooxygenase activity of the CYP76AH subfamily towards other terpene scaffolds” and Discussion, fifth paragraph.

The authors might consider adding to the phylogenetic tree shown in Figure 8 the demonstrated biochemical activities of the relevant enzymes, to help the reader understand the relationship between the phylogenetic analysis and the biochemistry.

Thank you the reviewer for this suggestion, we believe that can help the reader to better understand the activities landscape of these enzymes. The chemical structures of the main products of the CYP76AH enzymes mentioned in the text have been added to the phylogenetic tree.

In the NMR data, it would useful for the community if the actual NMR spectra are shown in addition to the tabulated chemical shifts shown in Table 1 and 2.

The NMR data generated during this work has been added to the manuscript as supplemental data, Figure 3—source data 1, PDF format.